

# Impact of humidity biases on light precipitation occurrence : observations versus simulations

Sophie Bastin[1], Philippe Drobinski[2], Marjolaine Chiriaco[1], Olivier Bock[3], Romain Roehrig[4], Clemente Gallardo[5], Dario Conte[6], Marta Dominguez-Alonso[5], Laurent Li[2], Piero Lionello[6,7], Ana C. Parracho[1,3]

[1] LATMOS/IPSL, UVSQ Université Paris-Saclay, UPMC Univ. Paris 06, CNRS, Guyancourt, France
[2] LMD/IPSL, UPMC Univ. Paris 06, Ecole Polytechnique, ENS, CNRS, Palaiseau, France

[3] IGN LAREG, Université Paris Diderot, Sorbonne Paris Cité, Paris, 75013, France
[4] CNRM/Meteo-France, Toulouse, France
[5] UCLM, Instituto de Ciencias Ambiantales, Toledo, Spain

[6] CMCC, Euro Mediterranean Center on Climage Change, 73100 Lecce, Italy
[7] DiSTeBA, University of Salento, 73100 Lecce, Italy

**Abstract.** This work uses a network of GPS stations over Europe from which a homogenised integrated water
vapor (IWV) dataset has been retrieved, completed with colocated temperature and precipitation measurements
over specific stations to i) estimate the biases of six regional climate models over Europe in terms of humidity ;
ii) understand their origins ; iii) and finally assess the impact of these biases on the frequency of occurrence of
precipitation. The evaluated simulations have been performed in the framework of HYMEX/Med-CORDEX
programs and cover the Mediterranean area and part of Europe at horizontal resolutions of 50 to 12 km.
The analysis shows that models tend to overestimate the low values of IWV and the use of the nudging
technique reduces the differences between GPS and simulated IWV. Results suggest that physics of models
mostly explain the mean biases, while dynamics affects the variability. The land surface/atmosphere exchanges
affect the estimation of IWV over most part of Europe, especially in summer. The limitations of the models to
represent these processes explain part of their baises in IWV. However, models correctly simulate the
dependance between IWV and temperature, and specifically the deviation that this relationship experiences
regarding the Clausius-Clapeyron law after a critical value of temperature ($T_{break}$). The high spatial variability of
$T_{break}$ indicates that it has a strong dependence on local processes which drive the local humidity sources. This
explains why the maximum values of IWV are not necessarily observed over warmer area, that are often dry
area.
Finally, it is shown over SIRTA observatory (near Paris) that the frequency of occurrence of light precipitation is
strongly conditioned by the biases in IWV and by the precision of the models to reproduce the distribution of
IWV as a function of the temperature. The results of the models indicate that a similar dependence occurs in
other areas of Europe, especially where precipitation has a predominantly convective character. According to the
observations, for each range of temperature, there is a critical value of IWV from which precipitation picks up.
The critical values and the probability to exceed them are simulated with a bias that depends on the model.



Those models which present too often light precipitation generally show lower critical values and higher probability to exceed them.



## 1 Introduction

Humidity plays a major role in the water and energy cycles due to its strong radiative effect associated with a positive feedback on climate (Randall et al., 2007) and its importance to control precipitation and particularly

extreme ones (Held and Soden, 2006 ; Neelin et al., 2009 ; Sahani et al., 2012). Trends and variability of humidity and precipitation are strongly correlated (Trenberth et al., 2003 ; Zhang et al., 2013) and several studies have revealed that the rate of increase in daily extreme precipitation is highly connected with the warming following the Clausius-Clapeyron (C-C) relation (Allen and Ingram, 2002; Pall et al., 2007; Kharin et al., 2007). This rate of precipitation is indeed affected by the humidity content of the atmosphere (Integrated water Vapor

(IWV)), which rises as climate warms (e.g Trenberth, 2011). Nevertheless, dynamical processes (O'Gorman and Schneider, 2009; Sugiyama et al., 2010; Singleton and Toumi, 2013; Muller, 2013 ; Drobinski et al., 2016a), lack of humidity sources leading to a decrease of relative humidity (Drobinski et al., 2016b), or low/high precipitation efficiency (Drobinski et al., 2016a ; Trenberth et al., 2003) can explain the deviation from C-C rate locally. Humidity variability at regional scale -and not only at the surface - thus needs to be assessed to better

anticipate the precipitation change, and more specifically the rate of heavy precipitation, that are not well estimated by global models (e.g Allan and Soden, 2008).

Another aspect that links IWV and precipitation concerns the triggering of precipitation and thus the frequency of occurrence of precipitation : Holloway and Neelin (2008) showed that precipitation over the tropical oceans is strongly sensitive to free-tropospheric humidity even more than surface humidity, and Neelin et al. (2009) and

Sahany et al. (2012) further conclude that there exists a threshold of IWV, which depends on the mean tropospheric temperature, over which precipitation starts to increase significantly. They also showed that this critical value of IWV does not correspond to the saturation value when temperature increases, i.e that at higher temperature, deep convection occurs at a lower value of relative humidity. This means that IWV is a relevant parameter to measure over long-term periods, at high temporal resolution and at the regional scale in order to

establish the relationship between IWV-precipitation and temperature and monitor its possible evolution. Models still have strong difficulties to simulate adequately the water cycle (Trenberth et al., 2003, Flato et al., 2013), and often presents the « too often too light precipitation » problem (e.g Sun et al., 2006 ; Panthou et al., 2016). A better knowledge of the IWV-precipitation relationship would be a help to better constrain models.

Up to now, very few long-term (> 15 years) and homogeneous datasets of water vapor measurements exist, even

less at sub-daily time scales. These datasets are necessary to understand the humidity variability at regional scales at different time scales. Besides, the co-location of such measurements with independent measurements of precipitation and vertical profiles of temperature provide a strong added-value for better climate understanding. Reanalyses are of course a good tool to have these three parameters co-located over long-term and at sub-daily time scales, however precipitation mostly relies on the model physics. Moreover, Flato et al. (2013) have shown

that even in reanalyses, the relationship between the IWV trend and the temperature trend presents differences between reanalyses and deviates from C-C over tropical oceans.

In this study, we make use of the same Global Positioning Sysem (GPS) IWV dataset as used by Parracho et al., (2018) which provides IWV measurements over a hundred of European sites. The GPS technique accurately measures IWV in all weather conditions including rainy situations, which is an important aspect for our study

(e.g Wang et al. 2007). GPS measurements have been successfully used to better understand atmospheric



processes at high resolution (Bastin et al., 2005, 2007 ; Bock et al., 2008 ; Champollion et al., 2009). Here we use a GPS IWV dataset to analyse the humidity biases in regional climate models over Europe at interannueal, seasonal and daily time scales and to better understand the source of errors of models.  We also use GPS IWV measurements co-located with precipitation and tropospheric temperature measurements from the SIRTA

observatory in France to consider the relationships between these parameters. Note they have not yet been considered outside the tropics. We compare observations and regional models output at the site level, and extend the analysis of models to other locations over Europe.

The paper is organised as follow : section 2 presents the observational datasets and the different simulations used in this study. Section 3 describes the methodology to compare observations and models. In section 4, the ability

of models to reproduce the mean value of humidity and its variability over Europe at different time scales is evaluated. The influence of dynamical and physical processes is discussed, and a special focus on the scaling of IWV with temperature is developed. In section 5, the issue of how much a bias in IWV can enhance the problem of 'too often too light precipitation' behavior of models is raised by considering the relationship between mean tropospheric temperature, IWV and precipitation in the different models and observations over the SIRTA

supersite in France. Then, the generalization of this relationship, by considering other stations over Europe is assessed. Finally, a conclusion is given in section 6.

## 2 Material

### 2.1 GPS IWV data

The GPS dataset used in this study is based on homogeneously reprocessed GPS delay data produced by the NASA's Jet Propulsion Laboratory in the framework of the first International GNSS Service (IGS) reprocessing campaign. The data cover the period from January 1995 to May 2011 and include more than 400 stations

globally. For the present study, the delay data were screened and converted into 6-hourly IWV estimates as described in Parracho et al. (2018). The dataset was restricted to the period from Januray 1995 to December 2008 including 95 GPS stations over Europe as shown in Fig. 1. Stations with less than 5 years of observations are not considered in the evaluation of models humidity bias in the present study.

### 2.2 Observations at SIRTA

This study also uses observations collected at the SIRTA atmospheric observatory, located 20-km South West of

Paris (2.2°E/48.7°N 160 m of altitude; black triangle on Fig. 1), from 2003 (Haeffelin et al. 2005). This observatory has collected many observations, which are now synthesized into the so-called "SIRTA-ReOBS dataset" described in Chiriaco et al. (2018) and used in Chiriaco et al. (2014) and Bastin et al. (2016). After many steps of data quality control and harmonization, the "SIRTA-ReOBS" file contains hourly averages of more than 50 variables at this site. The sample of data varies from one variable to another. Among these

variables, the IWV retrieved from GPS measurement since 2008 and the precipitation rate from a single raingauge over 2003-now are available. A regional-scale precipitation estimate, deduced from the measurements



of other Météo-France raingauges located around Paris area, is also provided.

The Météo-France COMEPHORE (« COmbinaison en vue de la Meilleure Estimation de la Precipitation HOraiRE ») product is used to allow a fairer intercomparison between models and observations than the single raingauge (Chen and Knutson, 2008): it is a hourly reanalysis of precipitation by merging radar data and

raingauges over France at 1km x1km resolution (more details in Fumiere et al., submitted ; see also Laurantin et al., ERAD 2012). From this product, we can have a better knowledge of the average precipitation rate over a model grid of 50 x 50 km or higher resolution. However, this product only covers the period 1997-2007 (at the time of this study) and is not concomitent with the IWV dataset over SIRTA. Despite this, a comparison has been made between the statistics of the different datasets when possible (see section 3.2).

The Météo-France radiosoundings launched twice a day from Trappes (near 00 and 12 UTC), 15 km to the west of SIRTA are also used to compute the mean tropospheric temperature (more details in section 3).

**2.3 Med-CORDEX simulations**

The list of regional climate models (RCMs) and details about the settings are given in Table 1. All the simulations use the 6-hourly European Center for Medium-Range Weather Forecast (ECMWF) reanalyses ERA-Interim (Berrisford et al., 2011) as RCM boundary conditions. They cover at least the period 1989–2008 as initially recommended in MED-CORDEX project (Ruti et al. 2015). For LMDZ, which is a global model with

regional zoom capability, temperature, wind speed and specific humidity are nudged towards the ERA-Interim fields outside the MED-CORDEX domain. It must be noted that the mesh of LMDZ is not regular within the zoom region and the resolution varies between 50 and 30 km. All other RCMs are forced at the boundaries using 3-dimensional re-analyses of wind, humidity, temperature or potential temperature and geopotential height. For CCLM, cloud ice and liquid water are additionally prescribed at the domain boundaries. The IPSL WRF

simulations use nudging at all scales within the domain for temperature, wind and humidity above the planetary boundary layer (Salameh et al. 2010 ; Omrani et al. 2013, 2015). The other models did not use nudging in the Med-CORDEX domain.

Simulations used here were produced from five models (ALADIN V5.2, Colin et al. 2010; CCLM, Rockel et al. 2008; WRF V3.1.1, Skamarock et al. 2008; LMDZ V4, Hourdin et al. 2006; PROMES, Dominguez et al., 2010 ;

2013). Five simulations were carried out with a horizontal resolution around 50 km (0.44° for most models) on the MED-CORDEX domain, and other three simulations was run with a higher resolution (0.11° or 0.18°). In the following the simulations are referred to by the name of the modeling group and resolution (see Table 1, first column). All the models provide daily values of IWV, precipitation, 2-m temperature, and temperature at 850, 500 and 200 hPa. The 6-hourly model outputs are used when available for comparison with GPS data.

**2.4 Others datasets**

The GPS observations are supplemented by the HadISD v.2.0.1.2016 subdaily dataset of surface parameters (e.g temperature, dew point temperature, wind, pressure - Dunn et al., 2012). It is global and based on the Integrated Surface Database (ISD) dataset from NOAA's National Climatic Data Center. Stations were selected on the

basis of their length of record and reporting frequency before they are passed through a suite of quality control





tests. It is a joint effort from the MetOffice Hadley Center and the National Center for Atmospheric research (NCAR).

## 3 Methods

### 3.1 Comparison between GPS dataset and model outputs

Each modeling group has provided a file containing the gridded IWV over the 1995-2008 period at daily or 6-hourly resolution. The IWV is either computed online or offline the model. The offline computation can introduce some errors due to vertical integration over the discretized vertical grid. For each model, we extracted the value of IWV from the gridded products at the closest grid point of GPS stations every 6 hours when available. The difference of altitude between the GPS station and the closest grid point is difficult to take into account and can introduce strong bias over complex terrain (Hagemann et al., 2003; Zhang and Wang, 2009). As 15 a consequence, the stations where the difference of altitude is higher than 500m were removed from the analysis. Note the number of stations that are removed depends on each model since the models do not use the same topography and the same projection. Then, a linear correction is applied on model outputs to reduce the bias due to orography : for each model and each month, we plotted the difference of the monthly averaged IWV values between the model and GPS as a function of the difference of altitude and we concluded that a linear correction 20 can be applied to take into account the difference of altitude. For each different month, the slope of the linear regression between these two differences is computed and we apply the corresponding correction to IWV values of the model. The values of the slope of the linear regression for each model and each month are indicated in Table 2. Various evaluation metrics (Table 3) has been computed with and without correction. The correction does not impact the variability scores (e.g diurnal cycle, interannual variability), but does affect the mean bias 25 (not always by an improvement) and slightly reduces the standard deviation of the difference (Table 3). It does not affect the ranking of performance between models.

Note that at SIRTA, the difference of altitude is weak and results are thus not impacted by this problem.

### 3.2 Comparison with SIRTA observations

i) Tropospheric temperature : to be as consistent as possible between model outputs and radiosoundings, the mean tropospheric temperature corresponds here to the daily average value of 2m-temperature, and temperatures at 850, 500 and 200 hPa. So we extracted the temperature values at these pressure levels for each radiosounding launched from Trappes (a few kilometers away from SIRTA) and computed the mean tropospheric temperature 35 as the average of these 4 values. Then, the daily mean is the average of the two daily radiosoundings. The same method is applied to ERA-Interim reanalysis to compute the mean tropospheric temperature over the other Euroepan sites. The impact of using ERA-Interim instead of radiosoudings data has been evaluated at SIRTA where both are available. Due to the larger of temperature bins considered in this study, results are not sensitive to the use of one or another.




ii) Precipitation: a first step consists in comparing the precipitation statistics obtained using the single raingauge located at SIRTA, from the closest grid point (1 km*1 km) of COMEPHORE over the SIRTA and an average of COMEPHORE over an area centered over the SIRTA and covering 2500 km$^2$ (i.e a 50-km resolution model grid point). Figure 2 shows the mean annual cycle of the frequency of occurrence of different light precipitation

regimes. The annual cycle is computed from 30-day means from January to December over the years 2004-2007, which is the common period of all datasets. The SIRTA raingauge and COMEPHORE product do not indicate the same frequency of occurrence of non-precipitating days, and very light precipitations, but they show similar frequency of occurrence for light precipitation and similar value of the 50$^{th}$ percentile of precipitation that is very low for both (zero, most of time). The difference between the frequency of occurrence of non–precipitating days

(higher with COMEPHORE at the closest grid point) and that of very light precipitation (higher with SIRTA raingauge) likely come from the coarse resolution of the coding of reflectivity data of the radars used at low levels which is a limiting factor for the precise estimation of precipitation at low rain rates (Laurentin et al., 2012). The difference is stronger in winter than in summer. The average over 50*50 km$^2$ grid cell shows a decrease in the number of non-precipitating days and an increase in the occurrence of very light precipitations : it

is expected since the probability that a system passes through a wider area is higher but the total precipitation averaged over a wide area is often much weaker as the strongest precipitation rates are generally localized. However, the value of 50$^{th}$ quantile of precipitation remains very low.

Table 4 presents an estimate of the frequencies of occurrence of non-precipitating days, very light precipitation and light precipitation for winter (julian day 1 to 100) and summer (julian day 151 to 251) when considering

either the common period of the two datasets (2004-2007) or the full period of each dataset (i.e 1997-2007 for COMEPHORE and 2003-2015 for SIRTA raingauge). The number of dry days increases for the two estimations from COMEPHORE when considering a longer period than the common period. For ReOBS, the statistics remain similar for the two different periods. However, when considering the most recent years only, that will be used in the next section (2008-2015), the number of dry days increases. The influence of the number of years,

the years considered and the products used to estimate these frequencies of occurrences is generally small but significant. Even though model errors are most of the time beyond this uncertainty, it has to be kept in mind in the following analysis.

Note that in Figure 2 and table 4, results from IPSL50 and IPSL20 are actually very similar which justifies the choice we made to use only the name of the institution for the model formulation.

### 3.3 Method to establish the relationship between IWV and precipitation as a function of tropospheric temperature

The objective is to characterize how precipitation depends on IWV for different range of mean tropospheric

temperature. Due to existence of gaps in the observations datasets which reduces the sample size, the number of temperature bins is different between observations and simulations : for observations, three bins of temperature are selected : the first one corresponds to values that range between the first percentile and 30th percentiles of temperature, the second bin to values between the 30th and 60th percentiles and the third one to values between 60th and 99th percentiles. The mean tropospheric temperature of each bin is indicated on the subplot (Fig.7a).

Then, in each bin of temperature, the 50th percentile of daily mean precipitation rates are sorted according





increasing values of daily mean IWV. IWV bins are then defined such that they contain an equal number of pairs of precipitation rates-IWV (40 samples for observations). The range of each IWV bin is thus not constant. This approach was preferred over that using IWV bins of equal width as it ensures a reasonable number of events across all bins, that is necessary to compute quantile values. The same methodology is applied to simulations,

with a few differences due to the greater sample size and to ensure that the number of samples in each IWV bin is close to that of observations : thus, four bins of tropospheric temperature are choosen (1th, 25th, 50th, 75th, 99th percentiles) instead of 3, and 50 pairs of the 50th percentile of precipitation (PR50) and daily mean value of IWV are used in each bin of IWV, instead of 40.

For each model and for observations, a critical value of IWV ($w_c$) is then determined for each different bin of

temperature by using a very simple algorithm. This one identifies the minimum value of IWV over which all precipitation rates are greater than 0.2 mm/day.

**4 Humidity biases in the Med-CORDEX simulations**

**4.1 Comparison with GPS dataset at regional scale**

Figure 1 indicates the mean values of IWV retrieved from GPS measurements in winter (Fig.1a) and in summer (Fig.1b). In winter, higher values are observed along the Atltantic and Mediterranean coasts while Central and Eastern Europe exhibit very low values of IWV. In summer, there are two different regimes: i) north of 45°N

showing a decrease of IWV values while going to Scandinavia, following the temperature gradient; ii) around the Mediterranean, the structure is more patchy with an alternance of low and high values. Most low values correspond to higher topography but not systematically.

Table 3 shows the mean bias and the root mean square error (RMSE) of ERA-interim and RCMs IWV, in comparison to GPS estimates. Either 6 hourly or daily datasets are used to compute these statistics and exactly

the same sampling is used between models and observations. It shows that the mean bias ranges from 0.4 to 1.0 kg.m$^{-2}$ except for CNRM50 and UCLM50 for which the biases are stronger (1.2 and 1.7 kg.m$^{-2}$ respectively). None of the model scores better than ERA-Interim despite their finer resolution. Twin simulations do not indicate major improvement when resolution is increased from 50 km to 20 or 12 km (CNRM, IPSL and CMCC).

The root mean square error indicates a large spread around observations for all models (~3.5-4.0 kg.m$^{-2}$), except IPSL which is even better than ERA-interim. The nudging towards ERA-interim used in this simulation likely explains its improved behaviour. The use of daily data instead of 6-hourly data does not modify the mean bias but it significantly decreases the RMSE, consistently with the work of Bock and Nuret (2009) on reanalyses.

To the first order, the IWV variability is dominated by the seasonal cycle (shown on Fig.1), which is

underestimated by models (not shown), and which explains part of the model RMSE : indeed, Figure 3 shows the percentage of simulated daily mean IWV values which overestimate GPS values at each station for the ensemble of the five models at 44-km resolution in winter and summer. Figures 3a and 3b present results without height-correction, while Figs.3c and 3d are done with height-corrected data. In winter, more than 70% of values are overestimated over most stations, with or without height corrections. In summer, this percentage decreases

appreciably in most stations and in almost half of them it reaches values below 50% (Fig.3b). The use of height



correction homogenizes the results in summer between stations and the very low or very high percentages do not appear anymore when this correction is applied. UCLM model is the moistest, especially in summer (not shown).

The IWV variability also comes from the interannual variability. For each month, we computed its anomaly by
substracting the average value of the month over all the years. We then computed the correlation between the anomalies of the GPS IWV estimates and those from the models. The minimum and maximum values of these correlations for each model when all stations are considered are indicated in Table 3. As indicated by these numbers that range between 0.78 and 0.99, the interannual variability is well captured by model, which is not surprising since this variability is mostly driven by large scale advection of air masses (all models use 6-hourly
ERA-Interim parameters as lateral boundary conditions). Note that the maximum correlation is very high for all models, while the minimum values are higher for ERAI, LMDZ and IPSL than for the three others. This is mostly due to a specific month (not shown) : in january 1996, three models have an anomaly very different from the observations and, as a consequence, the interannual correlation for january goes down.

A third time scale driving the IWV variability comes from the diurnal cycle. The fact that the RMSE is increased
when using 6-hourly data compared to daily data (Table 3) is explained by the difficulty of models to simulate the amplitude of the diurnal cycle (not shown). Most of values are overestimated during nighttime and underestimated in the afternoon. Models also tend to peak later in the afternoon, a result that has already been noticed over Scandinavia by Ning et al. (2013). These two aspects increase the deviation between models and observations at 6-hourly time scale.

To conclude, the RCM configuration allows a reasonable representation of the large scale advection of air masses by the models, which is an important driver of humidity within the RCM domains (see also Trenberth et al., 2005). This is further improved when the model is nudged towards reanalysis, as in the IPSL model (Tab.3). This good agreement is likely to be reduced when using GCM forcing at the RCM boundaries.

Nevertheless RCM errors are significant, especially at higher frequencies. They reflect the inability of RCMs to
fully capture the smaller scale processes associated with moisture sources and sinks (precipitation, mesoscale circulations, evaporation and evapotranspiration, clouds and microphysics). To better understand these errors, we assess the link between surface humidity and IWV at different time scales, since the boundary layer humidity strongly contributes to the IWV and is strongly impacted by the RCM representation of land surface/atmosphere interactions. Figure 4 displays the monthly mean values of IWV versus monthly mean values of 2-m specific
humidity (Q2) averaged over all stations where and when both IWV from GPS and Q2 from HadISD are available. Monthly means are computed if at least 60 concommitant (both IWV from GPS and Q2 from surface station) values are available (i.e about 2 values per day out of 4 possible). A total of 3238 months are obtained, spread over 42 different stations. The average number of stations per month is 19 with a maximum of 30 stations. Figure 4 shows that 2-m specific humidity is a very godd proxy for IWV at the monthly scale. All
models but IPSL have a similar relationship between the two variables to the observed one (slope of $2.4*10^3$ $kg.m^{-2}$). However, for a given surface humidity, IWV is generally overestimated by models, especially in the driest conditions. The IPSL model presents a different behavior : for low values of surface humidity, IPSL shows the same bias than the other models, while at higher Q2 values, its bias strongly increases, generating a different regression slope than observations. IPSL compensates its underestimation of surface humidity in summer (Bastin
et al., 2016) by a steeper slope. This compensation may be the result of the use of nudging or it can be due to a



deep boundary layer so that the total humidity contained within the boundary layer is similar to that of other models (not possible to check that). Since the nudging is only used above the boundary layer, and since most of humidity is contained within the boundary layer, there is little reason that the nudging totally explains this compensation. It means that for this model, Q2 is not a good proxy of total column humidity, even at monthly

scales. Note also that the spread between models is a bit higher in summer than in winter, most probably because of more active boundary layers and increased entrainment of humidity from the free troposphere at their top. These processes will also affect the link between surface humidity and IWV at scales shorter than a month.

Despite the strong correlation between the annual cycle of Q2 and those of IWV, Q2 is not necessarily a good proxy for IWV at other time scales or to tackle model biases (for instance IPSL bias for surface humidity is

strongly negative while it is weak and slightly positive for IWV). While in the wintertime, humidity variability mostly originates from the air mass advection, summertime variability is mainly affected by land-surface interactions and boundary layer processes. Several studies have shown the existence of a large spread in the representation of the surface fluxes and land-atmosphere coupling strength between models over Europe, due to the fact that Europe is a zone of transition between the regime of « energy limited » with low land-atmosphere

coupling strength and those of « soil moisture limited » with high land-atmosphere coupling strength. The difficulty to represent soil conditions and surface fluxes is then increased (Cheruy et al., 2015 ; Boe and Terray, 2014 ; Fischer et al., 2007 ; Knist et al., 2017). The interannual correlations between IWV and Q2 summertime anomalies are indicated for each model in Table 5, as averaged valeurs over all GPS stations and minium/maximum values across all GPS stations (an attempt was done for GPS IWV and HadISD Q2 but there

were too many missing values). For most stations, the correlation is higher than 0.5 with a mean value around 0.8 for the 6 models. The standard deviation of the difference between models is around 0.15, which reveals a good agreement between them. Some stations however indicate higher differences, as indicated by the minimum values that strongly differs between models. IPSL and LMD models present strong correlation at all stations (r > 0.52 and 0.43, respectively) while it is very weak at some stations for UCLM, CNRM and ERAI. It can be

explained by a stronger availability of surface humidity in summer, and then a weaker sensivity of IWV to the surface moisture availability. For the drier models (e.g IPSL), a dry interannual anomaly of soil moisture will have an impact on the surface evaporation and then on the IWV anomaly. On the other hand, over areas where the advection of air masses from the sea/ocean is a more important driver, the interannual variability of the large scale dynamics addects more strongly both IWV and surface humidity than surface processes. Depending on the

area and on the model, Q2 and IWV thus convey different but complemeantary infiormation about the model behaviour.

**4.2 Scaling of IWV with temperature**

A way to check the behavior of models is to consider the relationship between IWV and temperature. At global scale, the scaling of IWV with temperature is expected to follow the Clausius-Clapeyron (C-C) law, at a rate of about 6-7 %.°$C^{-1}$. At regional scale, this relationship deviates from C-C due to the strong influence of dynamics (air masses advection) and moisture availability (e.g Drobinski et al., 2016a). Figure 5a illustrates the scaling of IWV with temperature over the SIRTA station, which is representative of most stations over Europe. For the

lower temperature, the scaling follows C-C law. Above a critical temperature ($T_{break}$), IWV stops to increase at





this rate. This critical temperature, defined as the temperature when the slope of the relationship deviates from C-C, presents spatial variations that are displayed on Fig.5b for observations. For most stations, the critical value is between 15 and 18°C. It is higher for stations located around the Mediterranean sea, the Black Sea or at the eastern edge of the domain, in the Dniepr and Volga basins. The IWV value corresponding to this critical

temperature (which is close to the maximum IWV value reached at each station) is more variable (Fig.5c), due to a combinated effect of $T_{break}$ value, orography, and latitudinal differences.

Physically, $T_{break}$ corresponds to the value when the relative humidity significantly decreases due to a lack of humidity sources.

To determine the slope after $T_{break}$, we can approximate the expression of IWV :

$$IWV(T) = \int_{Z=0}^{Z=TOA} \rho_v(T,z)dz = -\frac{1}{g}\int_{P_S}^{P_{TOA}} Q(T,p)dP$$

with Q the specific humidity at altitude z, $\rho_v$ the density of water vapor in kg.m$^{-3}$, $P_s$ the surface pressure and $P_{TOA}$ the pressure at the Top Of Atmosphere (TOA).

As observed in average (e.g Ruzmaikin et al., 2014), the troposphere can be separated into two layers, one being the boundary layer (BL) and the other one the free troposphere (FT) : assuming constant humidity within the two

layers, IWV can be expressed as :

$$IWV(T) \approx \frac{1}{g}((P_s - P_{BL}) * Q_{BL}(T_{BL}) + (P_{BL} - P_{TOA}) * Q_{FT}(T_{FT}))$$

$P_s$ is about 1000 hPa, $P_{TOA}$ between 0 and 10 hPa, and $P_{BL}$ around 800-850 hPa, so that we can write :

$(P_s-P_{BL}) = \alpha \, (P_{BL}-P_{TOA}) \cong \alpha \, P_{BL}$ with $\alpha \cong 1/6-1/4$

For an air mass at temperature T and specific humidity Q, we can write $Q(T) = RH* Q^s(T)$, with exponent s for saturation and RH for relative humidity.

Which gives :

$$IWV(T) \approx \frac{1}{g}(\alpha \, P_{BL} * RH_{BL} * Q^s(T_{BL}) + P_{BL} * RH_{FT} * Q^s(T_{FT})) \qquad (1)$$

With our simplified profile in two layers, we have :

$Q^s(T_{FT}) \approx \alpha Q^s(T_{BL})$ if we consider that $T_{BL}$ is around 280 K and $T_{FT}$ around 260 K (for T<260, the value of T does not affect a lot the value of $Q^s$ since it has nearly reached its minimum value).

$RH_{FT}$ can approximately be considered as a constant, close to 30% in average over Europe (Ruzmaikin et al., 2014)

Equation (1) can thus be approximated by :

$$IWV(T) \approx \frac{\alpha}{g} P_{BL} \, (RH_{BL} + RH_{FT}) * Q^s(T_{BL})$$

And finally, by differentiation we obtain equation (2)

$$\frac{1}{IWV}\frac{\partial IWV}{\partial T} \approx \frac{1}{Q^s_{BL}}\frac{\partial Q^s_{BL}}{\partial T} + \frac{1}{RH_{BL}+RH_{FT}}\frac{\partial RH_{BL}}{\partial T} \qquad (2)$$



The first term of the right hand side (RHS) of Eq. (2) is the variation of water holding capacity of the atmosphere at temperature T and is thus Clausius-Clapeyron rate (~7%.C$^{\circ -1}$ at 25°C).

At temperature lower than $T_{break}$, RH is nearly constant, so that the second term of RHS of Eq. (2) is close to zero
and the slope of IWV as a function of temperature follows C-C. According to Eq. (2), the deviation of the slope from C-C after $T_{break}$ should correspond more or less to the rate of RH decrease, which depends on the considered station. Figure 5d shows RH and IWV as a function of T for SIRTA observations. RH starts to decrease significantly at $T_{b1}$~13°C, while IWV curve deflects at $T_{b2}$~16°C. Table 6 indicates the values of the left hand side term of Eq(2) and the second term of the right hand side of Eq. (2) before $T_{b1}$ (at 10°C) and after
$T_{b2}$ (at 20°C). In spite of the important approximations done to obtain Eq. (2), the RH variations in the boundary layer thus explain to the first order the scaling of IWV with T. The determination of humidity sources that explain the RH variability are thus crucial.

Figure 5a indicates that depiste the bias at low temperature, models generally capture the deviation from C-C, and the IWV maximum value that is reached at high temperature, except the UCLM model for which IWV
slightly continues to increasing with T. Note that the model slopes at low T are a bit lower than that of the C-C law and that derived from the observations. They are aslo often slightly different above the critical value, indicating some difficulties to simulate the relative humidity decrease. Figure 6a shows the model ensemble mean of the scaling at the SIRTA station. It confirms that the deviation from C-C exists but the transition is smoother in the models than in the observations which makes the estimate of $T_{break}$ more uncertain at this station.
However, some stations show more abrupt transitions (not shown). For the ensemble, due to the smooth transition, $T_{break}$ is thus defined as the temperature value when IWV stops to increase (i.e when it reaches its hiatus value). Figure 6b and c indicates the model ensemble mean value of $T_{break}$ and the corresponding $IWV_{break}$. The models capture the spatial pattern of $T_{break}$, especially the higher values close to the Mediterranean, the Black Sea and in the eastern edge of the domain, but tend to overestimate it compared to the observations
(Fig5b) but as already said, the uncertainty on the $T_{break}$ estimate is quite high. The maximum value of IWV is also generally well simulated by the models, except over the northern part of the domain where UCLM model does not always capture this break (as seen on Fig.5a), indicating an overestimation of relative humidity.

The link between the IWV and RH evolutions for the model ensemble is shown on Fig.6a. Once again, the transition from one regime to another is smoother than for observations (Fig.5d), but the decrease of RH starts
around the same range of temperature as for the transition of IWV-T relationship. Table 6 confirms that also in models RH variations in the boundary layer explain to a good degree the scaling of IWV with T.

In conclusion to this section models tend to overestimate low values of IWV. Although they generally well capture the IWV scaling with temperature, smal scale processes are likely to explain important SD when considering the differences with GPS IWV data. In the next section, we consider the impact of these humidity
biases in models on light precipitation occurrence.

**5 Impact of model biases on light precipitation**

**5.1 Over SIRTA**




Figure 7 displays the 50$^{th}$ percentile of precipitation (computed including days without precipitation) as a function of IWV for different bins of mean tropospheric temperature for both observations and the models (see section 3 for the details of the methodology). The observations are considered for the period 2008-2015 and the different models for the period 2001-2008 (i.e. the same number of years, despite the time shift). For all these

datasets, there exists a critical value of IWV, $w_c$, over which precipitation picks up. The values of these critical values are determined from Figure 7 following methodology indicated on section 3.3. The critical value depends on temperature (it increases with temperature) and on the models, as shown on Figure 8a. Three models (LMD, UCLM, CNRM) present lower critical values of IWV than observations at all temperatures. This means that in these models the precipitation begins in a drier atmosphere than that which begins to produce rain in the

observations. On the contrary, for CMCC and IPSL, the atmosphere needs to be as wet or wetter than what is observed to trigger precipitation. The first group of models are the same than those that present too often light precipitation and not enough days without precipitation (Fig.2).

The second reason of the high frequency of occurrence of very light precipitation in these models is that the probability of exceeding this critical value of IWV is strongly overestimated in the first group of models in

comparison with observations (Fig.8b). Statistically speaking, it means that the models that are too humid have a positive bias in light precipitation. For CMCC, though the critical value of IWV is correct, the probability to exceed it is too strong at low temperature, typically during winter, which explains why it rains too much during winter, while it is not the case during summer. IPSL model is dry both in terms of humidity and precipitation. It is also important to note that the underestimation of non-precipitating days for all models in winter is consistent

with the systematic overestimation of low values of IWV (Figs 3, 4, 5).

In addition to period 2001-2008 discussed above, we tested two other 8-year periods (1989-1996 and 1995-2002) and the entire period (1989-2008) to assess the influence of the considered period on the results. Figure S1 and Table 7 show that results are rather robust among models and periods though there exists some uncertainty in both $w_c$ and the probability to exceed it. The maximum relative variability in $w_c$ for one model is around 25%,

which is small compared to the maximum difference when considering all values from all models that is 58% for bin 1 (253 K), 73% for bin 2 (257 K), 63% for bin 3 (261 K), and 64% for bin 4 (264 K). The warming of the tropospheric temperature due to climage change is also visible on Fig. S1 and in Table 7 since for all models and for the three warmer temperature bins, the warmer temperature is obtained during the most recent period (2001-2008 ; i.e period 3). On the contrary, lower temperatures (first bin) tends to decrease, indicating a tendancy of

the distribution to become wider. We can note that due to the variability of the critical value of IWV and the variability of T inside each bin, the maximum value of $w_c$ inside each bin is not always obtained during the period of maximum temperature. For the first bin, 4 models out of 6 indicate a higher value of $w_c$ during the first period, which is the warmer for this bin. For the other three bins, the maximum temperature is obtained for all six models during the most recent period (100% of values) while this period gets only 39% (7 out of

18 cases) of maximum $w_c$ values, and the three other periods represent about 20% each.

### 5.2 Generalization

To have an idea of the models' behavior over other parts of Europe, several other stations are considered in this

section. Their locations are shown on Fig. 1b by black diamonds and details are given in Table 8. Except for



Marseille located in the south of France where the COMEPHORE product associated with GPS has been used, the model outputs are not compared with observations for the other stations. Figure 9 displays the occurrence of non-precipitating days for the different models and from COMEPHORE product at the station location when available (over France), $w_c$ values as a function of temperature and the percentage of IWV values that exceed $w_c$

as a function of temperature. The observed frequency of occurrence of non-precitating days is slightly higher in Marseille, located in the dry southern France, than in the northern part of France (SIRTA). Most of the models (except CNRM) overestimate this north-south gradient. For instance, for UCLM, the frequency is between 80 and 90% at Marseille, while it is around 15% at SIRTA. Several other characteristics of the models' behavior do not depend much on the station: CNRM always simulates less occurrence of non-precipitating days than other

models and simulates a rather flat annual cycle ; the annual cycle simulated by CMCC is always very intense, with much higher frequency of occurrence of non-precipitating days in summer than in winter, IPSL is always the model with the highest frequency of occurrence of non-precipitating days ; UCLM and LMDZ are between CNRM and IPSL, with a tendency to simulate many days with very light precipitation at the northern stations (Fig.2 and Fig.9g, m) but not at the southern ones (Fig.9a, d). In the eastern part of the domain (Fig.9j), the

annual cycle of precipitation simulated by models is a bit different than elsewhere with two drier periods in spring and fall. Also note that the station located in Central Europe (Germany) is the only one where the model's resolution impacts the results.

To relate these caracteristics to temperature and humidity, we reproduced the analysis done at SIRTA. The value of the critical value of IWV over which precipitation picks up is generally similar among models, despite

increasing dispersion with temperature. This value depends on the stations, indicating the influence of local specifities in the estimation of this relationship. For instance, UCLM model, which is the model with the most important difference between southern and northern stations for the annual cycle of dry days also indicates strong differences in the $w_c$ for a similar temperature, with a critical value around 25 kg.m$^{-2}$ in Marseille for a tropospheric temperature of 260K (Fig.9b), while it is ~12 kg.m$^{-2}$ in the Netherlands (Fig.9n). The nature of

precipitation, more or less convective, likely explains these differences. The probability to exceed the critical value is the most discriminant parameter between models and between seasons. Results are robust and confirm the importance of the relationship between temperature, IWV and light precipitation : for a given bin of temperature if the model is too humid (higher probability to exceed the threshold), it rains too often. The humidity bias thus strongly affects the low precipitation rates, more than the threshold of precipitation triggering.

**6 Conclusion**

This work uses GPS integrated water vapor measurements associated with temperature and precipitation

measurements to i) estimate the biases of six regional climate models over Europe in terms of humidity ; ii) understand their origins ; iii) and finally assess the impact of these biases on the occurrence of precipitation.

The first part of the study aimed at evaluating the mean bias and standard deviations of IWV in models compared to GPS measurements at interannual, seasonal, and daily time scales. An interesting result is that all models overestimate the lower values of IWV (nighttime, wintertime) at all stations. The spread among models

is increased during summertime. Our analysis suggests that the model physics mostly explain the mean biases,



while dynamics affects the variability. The use of nudging towards reanalyses thus improves the representation of the large scale advections of air masses and reduces the standard deviation of differences between GPS retrieved IWV and simulated ones. The land surface/atmosphere interactions are crucial in the estimation of IWV over most part of Europe, especially in summer, and explain part of the mean biases. However, the

relationship between IWV and temperature, that deviates from the Clausius-Clapeyron law after a critical value of temperature, is generally well captured by models. This critical temperature presents a spatial variability since it corresponds to the value when relative humidity starts to decrease. It is thus strongly dependent on local processes which drive the local humidity sources (from evaporation and advection). This explains why the maximum values of IWV are not necessarily observed over warmer areas, which often corresponds to dry areas,

where soil moisture limited regime is dominant.

The improvement in humidity representation may also help in the representation of precipitation distribution. Indeed, in the second part of this study, it is shown that the biases in IWV and most importantly IWV's distributions as a function of temperature strongly impact the occurrence of light precipitation over France, and most generally over areas where convection is the main process of precipitation triggering. For each range of

mean tropospheric temperature, there exists a critical value of IWV over which a pickup in precipitation occurs. This is observed and simulated by models, but the critical values and the probability to exceed them vary between models and observations. Models which present too often light precipitation generally show lower critical values and higher probability to exceed them. Thus, a better knowledge and representation of the triggering thresholds of precipitation and of their variablity should potentially help to improve the representation

of the whole precipitation distribution in models. The ensemble of simulations with implicit and explicit convection that will be performed in the framework of the Flagship Pilot Study Convective-Permitting Climate Simulation of CORDEX project will allow us to assess the sensitivity of precipitation triggering and distribution to the model resolution.

*Acknowledgements* : This work is a contribution to the HyMeX program (HYdrological cycle in The Mediterranean EXperiment) through INSU-MISTRALS support and the MEDCORDEX program (Coordinated Regional climate Downscaling EXperiment—Mediterranean region). This research has received funding from the French National Research Agency (ANR) project REMEMBER (grant ANR-12- SENV-001) and is a
contribution to the VEGA project through LEFE/INSU support and to the GNSS4SWEC COST action ES1206 through EU support. It was supported by the IPSL group for regional climate and environmental studies, with granted access to the HPC resources of GENCI/IDRIS (under allocation i2011010227), by SIRTA Working Group 'Climate studies' and by the national infrastucture ACTRIS-FR. SIRTA-ReOBS effort also benefited from the support of the L-IPSL funded by ANR under the "Programme d'Investissements d'Avenir (Grant ANR-
10-LABX-0018) and by the EUCLIPSE project funded by the European Commission under the Seventh Framework Program (Grant no 244067). To process the data, this study benefited from the IPSL mesocenter ESPRI facility which is supported by CNRS, UPMC, Labex L-IPSL, CNES and Ecole Polytechnique. We would like to acknowledge the SIRTA team for collecting data; Cindy Lebeaupin-Brossier and Marc Stefanon for providing simulation outputs; the CNES (Centre National d'Etudes Spatiales) for partially funded M.
Chiriaco research, Emmanuele Lombardi and ENEA for the Med-CORDEX database, Samuel Somot for his role in Med-CORDEX coordination, and Thomas Noel for its help in post-processing data. The work to carry out the simulations of the UCLM model was funded by the Spanish Ministry of Education and Science and the European Regional Development Fund, through grant CGL2007-66440-C04-02.

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





5    **Table 1 : List of models used in the study**

| Name in this paper | Model/resolution | Number of levels | Radiation scheme | Convective scheme | Microphysics scheme | Land surface scheme | PBL scheme | Land use |
|---|---|---|---|---|---|---|---|---|
| CMCC50 | CCLM4-8-19/MED44 | 45 | Ritter and Geleyn (1992) | Tiedtke (1989) | Doms et al., (2007) and Baldauf and Schulz (2004) | Doms et al. (2007) | Louis (1979) | GLC2000 |
| CMCC11 | CCLM4-8-19/MED11 | 45 | Ritter and Geleyn (1992) | Tiedtke (1989) | Doms et al., (2007) and Baldauf and Schulz (2004) | Doms et al. (2007) | Louis (1979) | GLC2000 |
| CNRM50 | ALADIN52/MED44 | 31 | Morcrette (1990) | Bougeault (1985) | Ricard and Royer (1993) and Smith (1990) | Noilhan and Planton (1989) and Noilhan and Mahfouf (1996) | Ricard and Royer (1993) | GLC2000 |
| CNRM11 | ALADIN52/MED11 | 31 | Morcrette (1990) | Bougeault (1985) | Ricard and Royer (1993) and Smith (1990) | Noilhan and Planton (1989) and Noilhan and Mahfouf (1996) | Ricard and Royer (1993) | GLC2000 |
| IPSL50 | WRF311/MED44 | 28 | Mlawer et al. (1997) and Dudhia (1989) | Kain (2004) | Hong et al. (2004) | Smirnova et al., (1997) | Noh et al. (2003) | USGS |
| IPSL20 | WRF311/MED18 | 28 | Mlawer et al. (1997) and Dudhia (1989) | Kain (2004) | Hong et al. (2004) | Smirnova et al., (1997) | Noh et al. (2003) | USGS |
| LMD | LMDZ4NEMO8/MED44 | 19 | Fouquart and Bonnel (1980) Morcrette et al. (1986) | Emmanuel (1993) | Li (1999) and Hourdin et al. (2006) | Krinner et al. (2005) | Louis (1979) | None |
| UCLM50 | PROMES/MED44 | 37 | Mlawer et al. (1997), Morcrette et al. (2008) | Kain (2004) | Hong et al. (2004) | Krinner et al. (2005) | Cuxart et al. (2000) | IGBP |





**Table 2 : values of linear regression slopes obtained for each model/each month when plotting IWVmod-IWVobs as a function of difference of altitudes between model and GPS at each station. It is expressed in $10^{-3}$ kg.m$^{-3}$**

|  | CMCC50 | CNRM50 | IPSL50 | LMD | UCLM50 |
|---|---|---|---|---|---|
| *January* | -4.1 | -5.8 | -4.8 | -4.5 | -4.7 |
| *February* | -4.1 | -5.3 | -4.7 | -4.5 | -4.7 |
| *March* | -4.6 | -5.4 | -5.1 | -4.9 | -4.9 |
| *April* | -5.7 | -5.6 | -5.4 | -5.8 | -5.8 |
| *May* | -7.0 | -7.0 | -6.6 | -7.0 | -6.8 |
| *June* | -8.3 | -7.6 | -8.2 | -7.5 | -7.9 |
| *July* | -9.8 | -8.9 | -8.5 | -8.2 | -8.6 |
| *August* | -11.2 | -8.2 | -9.0 | -8.6 | -8.6 |
| *September* | -8.4 | -7.6 | -8.1 | -7.2 | -8.1 |
| *october* | -8.0 | -7.7 | -7.9 | -7.2 | -7.7 |
| *November* | -5.4 | -6.2 | -5.8 | -5.7 | -6.0 |
| *December* | -4.4 | -5.8 | -5.0 | -4.9 | -5.2 |





**Table 3: Mean bias and standard deviation (SD) in kg.m$^{-2}$ of the differences between models and GPS observations using 6-hourly or daily time resolution. The second number is the one obtained with the correction due to altitude difference. The rightmost column indicates the minimum and maximum values of the correlation of the interannual variability of monthly GPS anomalies and model anomalies (one correlation computed by month).**

| | Bias (6 hourly) | | Bias (daily) | | SD of difference (6 hourly) | | SD of difference (daily) | | Interannual correlation (min/max) |
|---|---|---|---|---|---|---|---|---|---|
| ERAI | 0.42 | 0.61 | 0.42 | 0.61 | 2.15 | 1.77 | 1.77 | 1.30 | 0.92/0.99 |
| LMDZ | 0.59 | 0.73 | 0.59 | 0.74 | 3.98 | 3.81 | 3.31 | 3.11 | 0.91/0.99 |
| IPSL50 | 0.82 | 0.80 | 0.82 | 0.80 | 2.01 | 1.82 | 1.62 | 1.38 | 0.91/0.99 |
| CNRM50 | 1.17 | 1.10 | 1.17 | 1.10 | 3.85 | 3.77 | 3.21 | 3.11 | 0.77/0.98 |
| CMCC50 | 0.45 | 0.57 | 0.44 | 0.57 | 4.08 | 3.92 | 3.36 | 3.16 | 0.78/0.96 |
| UCLM50 | 1.68 | 1.57 | 1.68 | 1.57 | 4.27 | 4.17 | 3.60 | 3.47 | 0.80/0.98 |
| IPSL20 | 0.95 | 0.88 | 0.94 | 0.88 | 2.22 | 1.99 | 1.81 | 1.53 | |
| CNRM11 | 0.97 | 0.93 | 0.97 | 0.93 | 3.63 | 3.62 | 2.97 | 2.96 | |
| CMCC11 | 0.71 | 0.59 | 0.71 | 0.58 | 4.18 | 4.14 | 3.40 | 3.35 | |





**Table 4 : Occurrence (%) of non –precipitating days (first number), very light precipitation (second number) and light precipitation (third number) for different datasets (columns) computed over different years (rows) for two different periods of the year: W is for winter (Julian day from 1 to 100) and S is for summer (julian day from 151 to 251).**

| | | REOBS | COM 1pt | COM maille | LMD50 | UCLM50 | CMCC50 | CNRM50 | IPSL50 | IPSL20 |
|---|---|---|---|---|---|---|---|---|---|---|
| 2004-2007 | W | 45/24/8 | 56/7/8 | 34/24/9 | 20/43/11 | 13/41/18 | 5/55/13 | 2/37/18 | 33/39/8 | 34/34/11 |
| | S | 58/16/4 | 67/10/5 | 47/27/6 | 29/41/9 | 18/38/15 | 41/40/5 | 2/43/11 | 59/21/4 | 59/20/5 |
| 1989-1996 | W | - | - | - | 24/40/11 | 12/41/17 | 7/56/10 | 1/42/14 | 35/38/7 | 35/36/8 |
| | S | - | - | - | 36/42/6 | 22/38/16 | 49/35/4 | 0/48/8 | 65/20/4 | 66/18/5 |
| 1995-2002 | W | - | - | - | 23/38/10 | 23/34/14 | 7/53/11 | 1/42/14 | 36/35/8 | 33/35/9 |
| | S | - | - | - | 33/41/7 | 24/36/15 | 45/37/4 | 1/45/9 | 62/20/4 | 64/17/5 |
| 2001-2008 | W | - | - | - | 23/28/11 | 14/37/17 | 6/54/13 | 2/37/16 | 35/37/8 | 35/34/10 |
| | S | - | - | - | 31/40/7 | 16/41/15 | 42/38/5 | 1/44/10 | 62/19/4 | 63/17/5 |
| 1989-2008 | W | - | - | - | 24/38/11 | 17/38/16 | 7/55/11 | 1/41/15 | 36/37/8 | 35/35/9 |
| | S | - | - | - | 33/42/6 | 22/38/15 | 45/37/4 | 0/46/9 | 62/21/4 | 63/18/5 |
| 1997-2007 | W | - | 62/6/7 | 44/22/8 | 24/38/11 | 22/36/16 | 7/54/12 | 1/40/15 | 36/37/8 | 35/35/10 |
| | S | - | 69/8/5 | 53/21/6 | 31/40/7 | 23/37/14 | 44/37/4 | 1/45/9 | 61/21/4 | 61/18/5 |
| 2008-2015 | W | 53/17/6 | - | - | - | - | - | - | - | - |
| | S | 63/13/5 | - | - | - | - | - | - | - | - |
| 2003-2015 | W | 47/18/6 | | | | | | | | |
| | S | 57/13/4 | | | | | | | | |





**Table 5 : Summertime interannual correlation between IWV and Q2 at GPS stations. In bold, the mean value for each model, followed by min and max values. The other values in the table corresponds to the standard deviation of the difference of correlation between two models.**

|  | CMCC50 | CNRM50 | IPSL50 | LMD | UCLM50 | ERAI |
|---|---|---|---|---|---|---|
| CMCC50 | **0.81/0.30/0.96** | 0.18 | 0.14 | 0.13 | 0.18 | 0.16 |
| CNRM50 | 0.18 | **0.76/0.05/0.97** | 0.16 | 0.15 | 0.19 | 0.20 |
| IPSL50 | 0.14 | 0.16 | **0.81/0.52/0.98** | 0.11 | 0.15 | 0.20 |
| LMD | 0.13 | 0.15 | 0.11 | **0.83/0.43/0.96** | 0.13 | 0.16 |
| UCLM50 | 0.18 | 0.19 | 0.15 | 0.13 | **0.76/0.24/0.96** | 0.21 |
| ERAI | 0.16 | 0.20 | 0.20 | 0.16 | 0.21 | **0.76/0.13/0.97** |



**Table 6 : Values of the left hand side (LHS), 2$^{nd}$ term of the right hand side (RHS) and total RHS of Eq. (2) computed for the SIRTA site for two different temperatures (10°C and 20°C), according to Fig.5d for observations and Fig.6a for models.**

| | 10°C | | | 20°C | | |
|---|---|---|---|---|---|---|
| | LHS | RHS, term 2 | RHS total | LHS | RHS, term2 | RHS total |
| OBS | *IWV~15 kg.m$^{-2}$* *Slope = 1 kg.m-2. C$^{°-1}$* ***Value = 6.6%. C$^{°-1}$*** | *RH~85%+30%* *Slope = -0.8%.C$^{°-1}$* *Value = -0.7%. C$^{°-1}$* | *6.6-0.7=**5.9 %. C$^{°-1}$*** | *IWV~23 kg.m$^{-2}$* *Slope = 0.5 kg.m-2. C$^{°-1}$* ***Value = 2.2%. C$^{°-1}$*** | *RH~60%+30%* *Slope = -3%.C$^{°-1}$* *Value = -3.3%. C$^{°-1}$* | *6.2-3.3 =* ***2.9 %. C$^{°-1}$*** |
| IPSL | *IWV~15 kg.m$^{-2}$* *Slope = 0.8 kg.m-2. C$^{°-1}$* ***Value = 5.2 %. C$^{°-1}$*** | *RH~80%+30%* *Slope = -0.8%.C$^{°-1}$* *Value = -0.7%. C$^{°-1}$* | *6.6-0.7=**5.9 %. C$^{°-1}$*** | *IWV~22 kg.m$^{-2}$* *Slope = 0.2 kg.m-2. C$^{°-1}$* ***Value = 1.1%. C$^{°-1}$*** | *RH~47%+30%* *Slope = -3%.C$^{°-1}$* *Value = -3.9%. C$^{°-1}$* | *6.2-3.9 =* ***2.3 %. C$^{°-1}$*** |
| Model ensemble | *IWV~16.5 kg.m$^{-2}$* *Slope = 1 kg.m-2. C$^{°-1}$* ***Value = 6.1 %. C$^{°-1}$*** | *RH~90%+30%* *Slope = -0.6%.C$^{°-1}$* *Value = -0.5%. C$^{°-1}$* | *6.6-0.5=**6.1 %. C$^{°-1}$*** | *IWV~25 kg.m$^{-2}$* *Slope = 0.4 kg.m-2. C$^{°-1}$* ***Value = 1.6%. C$^{°-1}$*** | *RH~70%+30%* *Slope = -4%.C$^{°-1}$* *Value = -4%. C$^{°-1}$* | *6.2-4 = **2.2 %. C$^{°-1}$*** |





Table 7 : Period for which the maximum value of the temperature bin (T), the critical value of IWV ($w_c$) . Period 1 is for 1989-1996 ; P2 for 1995-2002 ; P3 for 2001-2008. The second number in the column of Wc corresponds to the maximum relative variability of IWV computed as ((max(Wc)-min(Wc))/mean(Wc)) for each temperature bin and each model

|  | Bin1-253 K | | Bin2-257K | | Bin3-261K | | Bin4-264K | |
|---|---|---|---|---|---|---|---|---|
|  | T | Wc | T | Wc | T | Wc | T | Wc |
| IPSL50 | P1 | P1/7% | P3 | P3/10% | P3 | P2/11% | P3 | P1/9% |
| CMCC50 | P1 | P3/15% | P3 | P2/7% | P3 | P1/15% | P3 | P1/6% |
| LMD | P1 | P1/22% | P3 | P3/5% | P3 | P3/17% | P3 | P3/19% |
| CNRM50 | P1 | P1/20% | P3 | P3/7% | P3 | P3/8% | P3 | P2/16% |
| UCLM50 | P1 | P1/4% | P3 | P2/11% | P3 | P3/6% | P3 | P3/24% |



Table 8 : Latitude/longitude/altitude of the closest grid point of each model for the GPS stations used.

|  | LMD50 | CNRM50 | UCLM50 | CMCC50 | IPSL50 | IPSL20 |
|---|---|---|---|---|---|---|
| SIRTA 48.7/2.2/156 | 48.7/2.3/99 | 48.6/2.1/125 | 48.6/2.5/84 | 48.5/2.1/123 | 48.7/2.6/103 | 48.7/2.1/116 |
| MARSEILLE 43.3/5.4/12 | 43.3/5.4/132 | 43.5/5.1/35 | 43.1/5.6/0 | 43.2/5.5/112 | 43.1/5.6/61 | 43.3/5.4/130 |
| MADRID 40.4/-4.2/777 | 40.3/-4.4/731 | 40.5/-4.2/1130 | 40.4/-4.2/759 | 40.5/-4.5/922 | 40.4/-4.2/750 | 40.4/-4.3/840 |
| DRESDE 51.0/13.0/160 | 51.1/13.7/228 | 51.1/13.8/177 | 51.1/13.9/256 | 50.9/14.0/277 | 51.1/14.0/273 | 51.0/13.8/268 |
| KOOTWIJK 52.2/5.8/53 | 52.0/5.8/17 | 52.2/5.8/32 | 52.2/5.5/15 | 52.1/5.9/22 | 52.1/6.1/13 | - |
| POLTAVA 49.6/34.5/160 | 49.6/34.6/125 | 49.7/34.5/145 | 49.8/34.6/132 | 49.8/34.6/132 | 49.4/34.8/118 | 49.6/34.4/130 |





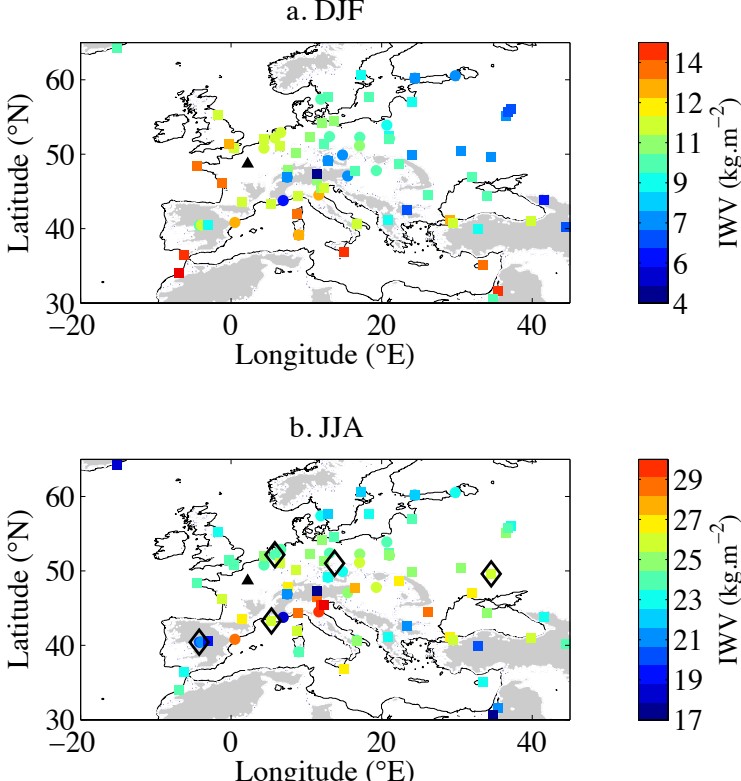

Figure 1: Mean values of IWV in winter (a) and summer (b) retrieved from GPS network. Note that color scales are different between winter and summer. Circles indicate stations with more than 10 years of observations between 1995 and 2008 and squares indicate stations with 5 to 10 years of observations. SIRTA observatory is shown by the black triangle. Black diamonds indicate the location of stations considered in section 5.2. Topography higher than 500 m above sea level is shaded in grey.





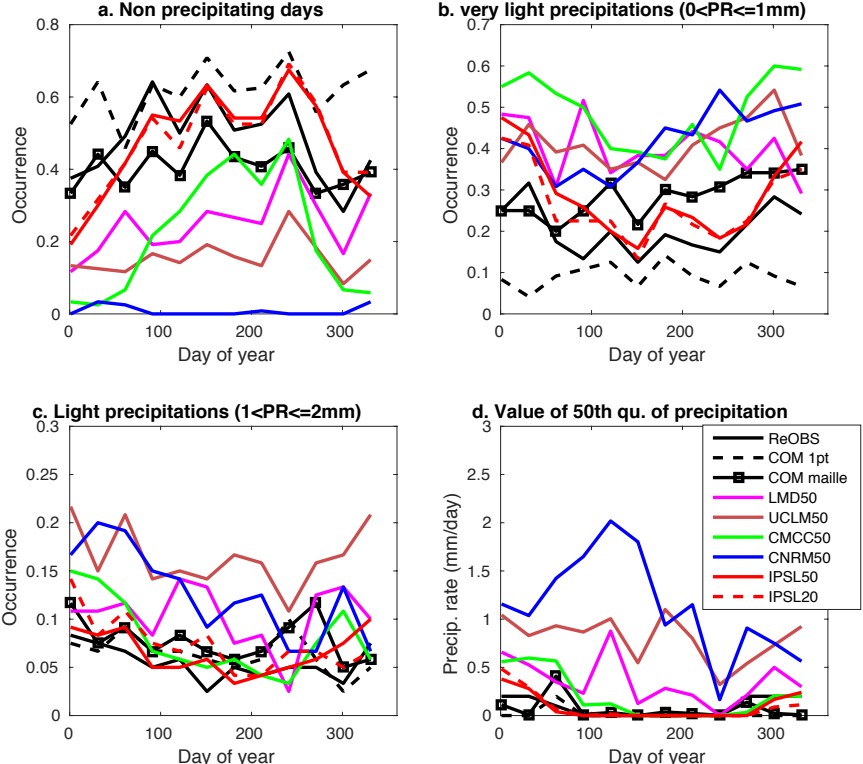

Figure 2 : Occurrence for 30-day periods from January to December over the years 2004-2007 of a. Non precipitating days ; b. precipitation rates between 0 and 1 mm/day ; c. precipitation rates between 1 and 2 mm/day. d. Value of the 50th quantile of precipitation (including non-precipitating days). Each color corresponds to a different model at 50 km resolution (see legend for details). Black line is for observations at SIRTA supersite. Dashed black line is for COMEPHORE product at the closest grid point of SIRTA and black line with squares is for COMEPHORE averaged over a square of 50*50 km2 around SIRTA.



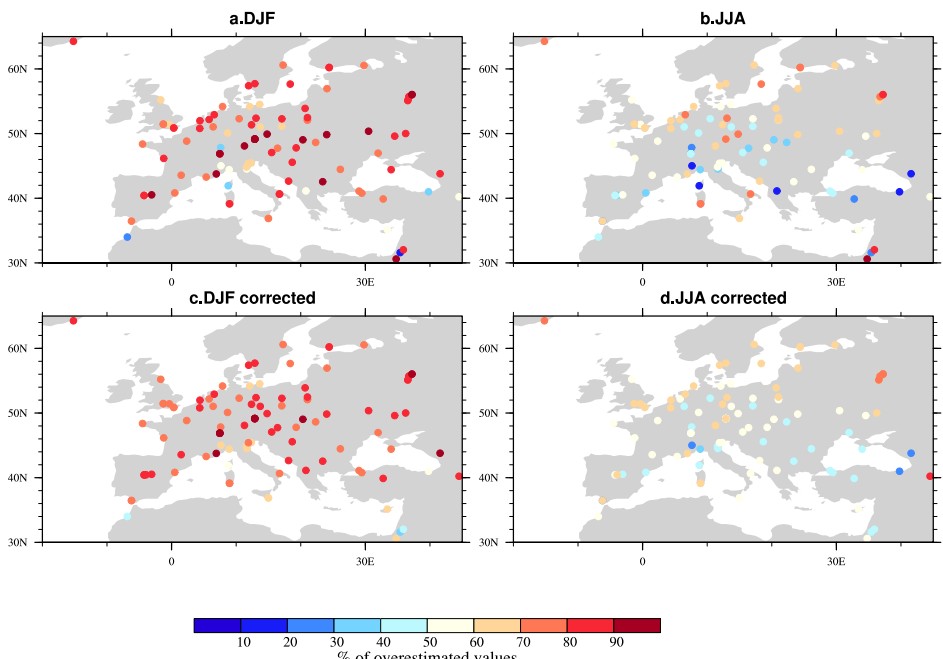

Fig. 3: Percentage of simulated daily mean IWV values which overestimate the GPS ones a)
in winter and b) in summer. Simulated values are taken from 5 models at 44 km resolution
(LMDZ, IPSL50, CNRM50, CMCC50, UCLM50).  c) and d) same as a) and b) but for
height-corrected values.





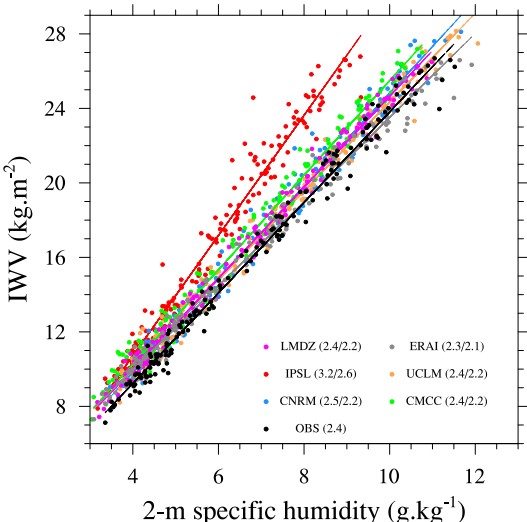

Fig. 4: Monthly values of IWV as a function of monthly values of 2-m specific humidity (Q2) averaged over all stations where and when both IWV from GPS and Q2 from HadISD are available (Monthly mean are computed if at least 60 co-existing values exist (i.e about 2 values per day over 4 possible). A total of 3238 months are obtained, spread over 42 different stations. The average number of stations per month is 19 with a maximum of 30 stations). The color of circles corresponding to each model is indicated on the legend. The first number indicates the slope of the regression obtained when considering the same months than observations at each station, while the second number is the slope of the regression when considering all months at all grid points (only models).




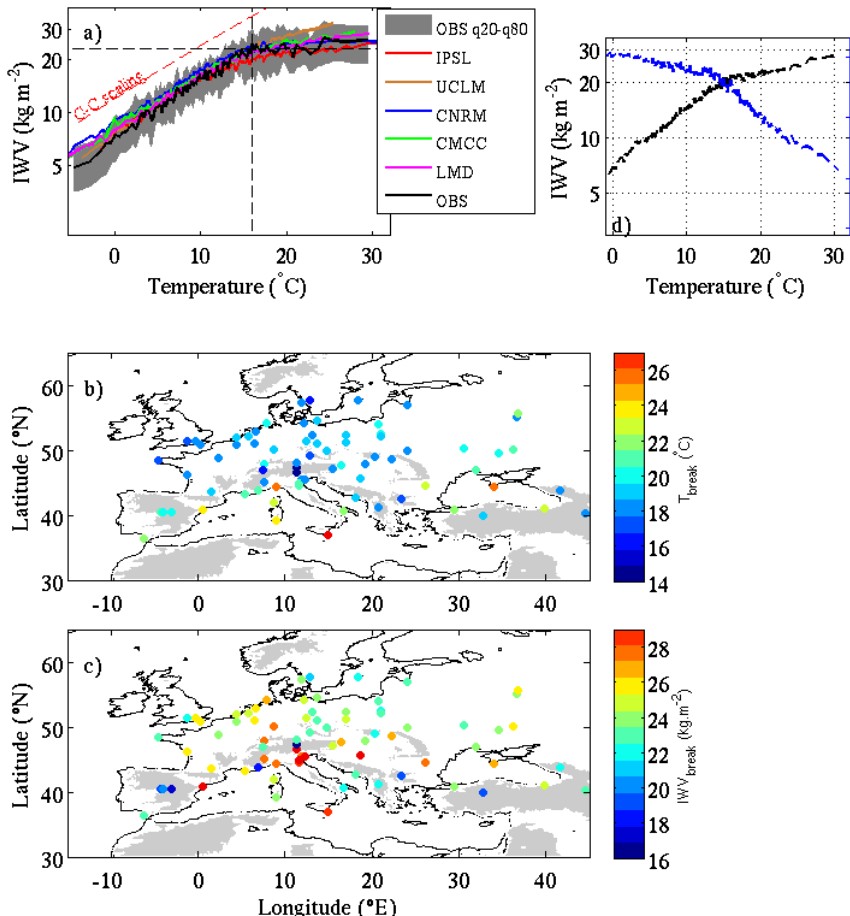

Fig. 5: **(a)** IWV – T(2m) relationship at SIRTA station from observations and models. Median values are plotted for observations and models. The grey band represents the interval between the 20th and 80th quantiles for observations. The black vertical dashed line shows $T_{break}$, the black horizontal dashed line shows $IWV_{break}$, and the red slant dashed line shows C-C scaling **(b)** Map of $T_{break}$ at all the GPS stations. **(c)** Map of $IWV_{break}$ at all the GPS stations. **(d)** Median values of IWV and RH(2m) as a function of T(2m) at SIRTA from observations only.





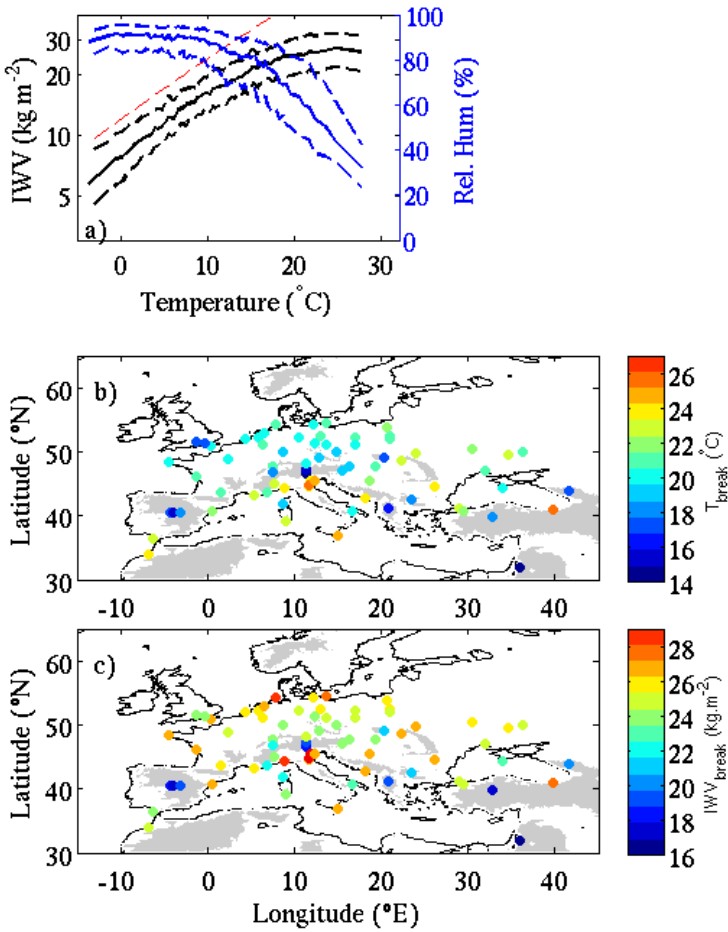

Figure 6 : a) Scaling of IWV (black) and RH (blue) with temperature for the model ensemble at SIRTA station. Solid lines are for quantile 50 and dashed lines are for quantiles 20 and 80 of the distributions. b) and c) As for Fig5 but for the model ensemble.





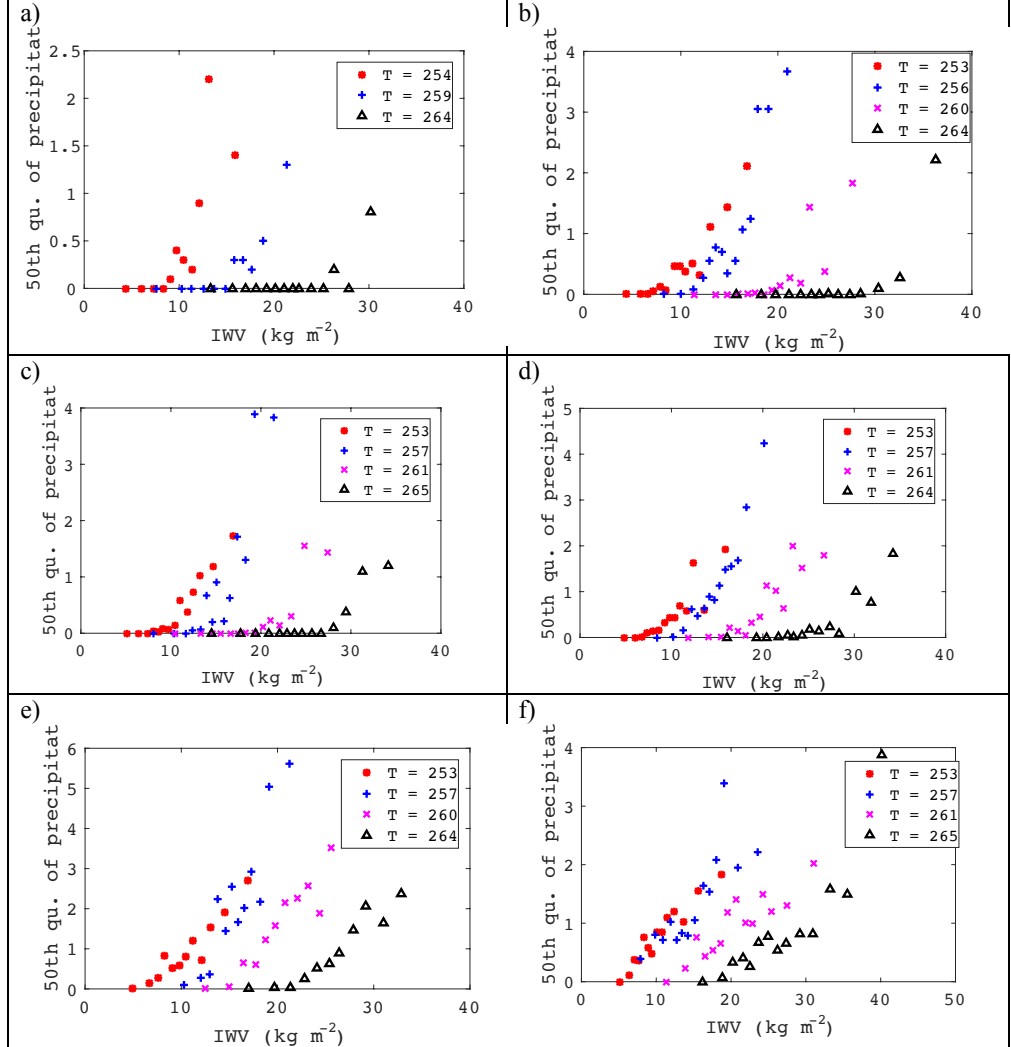

Figure 7 : 50th quantile of precipitation as a function of IWV for different bins of tropospheric temperature (3 bins for observations and 4 bins for models) at latitude 48.7°N and longitude 2.2°N (SIRTA). **(a)** SIRTA-ReOBS observations. **(b)** CMCC50. **(c)** IPSL50. **(d)** LMD50. **(e)** CNRM50. **(f)** UCLM50. Observations are analysed for the period 2008-2015, and models for the period 2001-2008. The values of T indicated in the legends correspond to the mean values in the bins that have been choosen as indicated in the text.




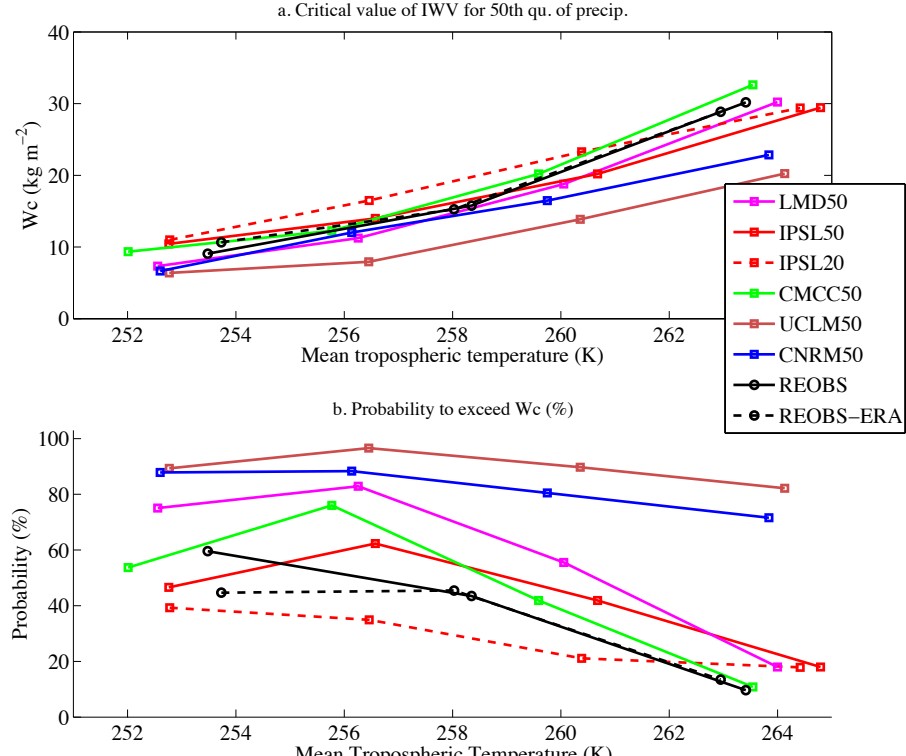

Figure 8 : **(a)** Critical value of IWV at latitude 48.7°N and longitude 2.8°E over which 50th
quantile precipitation significantly increases as a function of the mean tropospheric
temperature. Each color corresponds to a different model (see legend for details). The period
considered is 2001-2008. Solid black line is for observations from the SIRTA-ReOBS dataset
between 2008 and 2015 and dashed black line is for REOBS with ERA-Interim tropospheric
temperature. **(b)** Probability for IWV to exceed the critical value for each dataset.





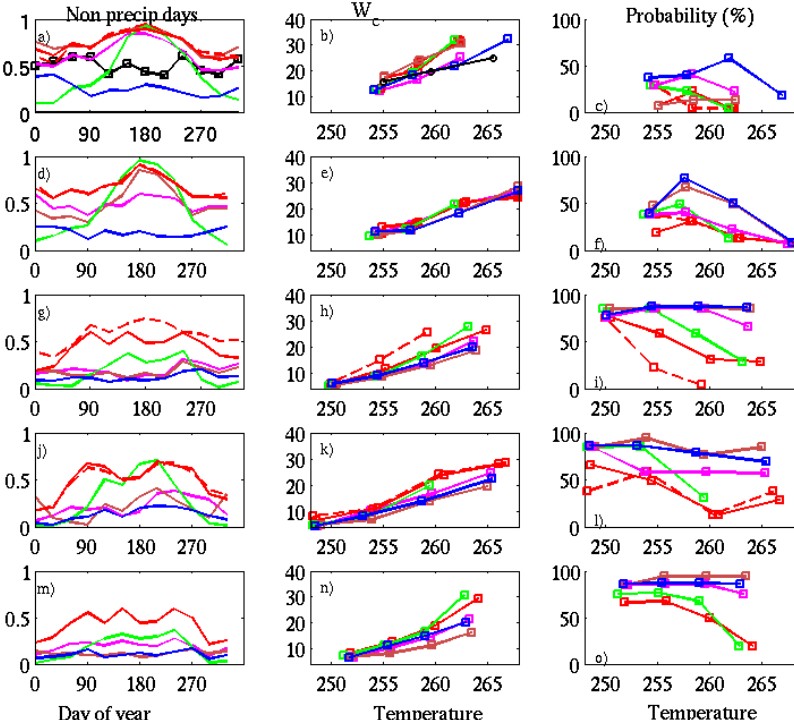

Figure 9 : first column : Annual cycle of occurrence of non-precipitating days for the different models and COMEPHORE in black in panel a (same legend than Fig.8 for models). Second column : Value of $w_c$ as a function of tropospheric temperature. Third column : Probability to exceed $w_c$ value. First row is for the station located in southern France (Marseille), second row for the one located in Central Spain (Madrid), third row for station in eastern Germany (Dresde), fourth row in Ukraine, and fifth row in Netherlands (see table 8 and Fig.1 for details on the locations of the stations).