# Peer review of "Impact of humidity biases on light precipitation occurrence"

_Atmospheric Chemistry and Physics, 2018_

## Referee Comment (RC1) · J.J. Gómez-Navarro (Referee) · 18 Jul 2018

**1 Abstract**

This study evaluates the performance of a number of regional climate models to re-produce humidity and precipitation. The emphasis of the article lies on the evaluation of Integrated Water Vapor (IWV), as well as its relation with temperature and finally with precipitation. It is found that models tend to overestimate the lower values of IWV, which is closely related to the "drizzling effect", i.e. too often too low precipitation.

**2 General comments**

I find this an interesting piece of work. The authors have made an effort to collect and put in comparable terms an heterogeneous set of data, from models to different observed variables at different locations and with different temporal availability. The use of an ensemble of simulations is a particularly good choice, as it provides robustness in the findings. I also like the fact that the authors do not just compare models with observations in plain terms, but they describe a simple two-layer atmosphere model that allows them to modelise the relationship between IWV and relative humidity with temperature, which allows them to gain insight on the sources of model biases. The text is well written and is easy to read, and the conclusions follow from the analysis carried out. Therefore I have only found very minor issues that the authors might want to consider.

**3 Specific comments & Technical corrections**

1. Pag. 6, line 23: has been → have been

2. Pag. 7, line 1: consists in → consists of

3. Pag. 7, line 34: Due to the existence of gaps in the observational dataset, which reduces...

4. Pag. 10, line 18: valeurs → values?

5. Pag. 11, line 20: I'd suggest that $s$, in $Q_s(T)$ to be called subscript, not exponent. An exponent is something else

6. Pag 10, line 28: Why do we have that $Q_s(T_{FT}) \approx \alpha Q_s(T_{BL})$? It is not obvious to me.
7. Pag. 12, line 8: What is $T_{b1}$ and $T_{b2}$?

8. Pag. 12, line 33: What is SD?

9. Figure 5: Should the right panel in the first row be labelled d? Further, that panel has particularly low resolution and generally lower presentation quality than, let's say, Figure 4

10. Figure 6: The same applies here. The first panel has bad quality and different aesthetics. I'd advice to follow the style followed to produce Fig. 4

11. Figure 7: the lines surrounding the panels are partly hidden by them and produce an ugly effect that should be avoided in the final version of the manuscript

12. Figure 9: The panels could be larger to take better advantage of the available space. Unlike in the other two, the third column has no right and top axis. The labels in the first row overlap with the axes. The low resolution issue applies here as well. There is no legend.

---

## Referee Comment (RC2) · Anonymous Referee #2 · 25 Sep 2018

The authors make use of integrated water vapour (IWV) and precipitation from a collection of observations (GPS stations, radio sounding, and combined radar/rain gauge data) in order to evaluate regional climate models operated on various grids (the grid spacing ranges from 0.11° to 0.44°) in a climatological manner (the periods of evaluation cover multiple years to decades). The climate models' simulations are driven by ERA-Interim and participated in the Med-CORDEX initiative. For this purpose, the authors develop a conceptual model that connects IWV, temperature, and precipitation (including Clausius-Clapeyron scaling and deviations from it) that helps to interpret the detected model biases and gives insights in the complexity of precipitation generating processes. From their analyses the authors conclude: 1) all models overestimate

lower values of IWV with an increasing spread among the models during summertime 2) mean biases are mostly explained by model physics (land surface/atmosphere interactions) while dynamics affect the variability 3) the IWV/temperature relationship (that deviates from the Clausius-Clapeyron law) is generally well represented by the models 4) biases in the frequency of occurrence in precipitation can be explained by a higher probability of exceedance of a critical value for IWV (that in turn depends on temperature)

General Comments

There is an endless number of evaluation papers for regional climate models (RCMs) that content themselves with showing biases, but there are only a few papers that deal with the sources of such biases and their underlying processes. The presented manuscript could be one of those rare papers. In addition, the manuscript elaborates on (even if just in a speculative manner) some aspects of the question, how climate change may affect the water cycle. The authors have provided a very interesting and innovative analysis that should be published as soon as possible. However, there are two methodological weaknesses that should to be clarified first, because they may affect the conclusions concerning the interpretation of biases and the precipitation/IWV function (Figure 7) drawn: (1) Comparability between reference (observational) and modelled data The authors make use of RCM data (including re-analysis data ERA-Interim) on various grids and compare it with data from stations (point data) by means of the nearest neighbouring method (cf. page 6, line 12). This has two implications: a) In such a comparison a coarser resolved model is more penalised than a model with a higher resolution, because it has a smaller spatial variability per construction. The coarser resolved model "sees" processes that are resolved by the model with a higher resolution only as "sub-grid scale effects". As a consequence, a judgement of biases drawn from models on their original grid can be misleading. In order to achieve comparability throughout models of different resolution, modelled and observed data is usually remapped onto a common coarsely resolved grid before the analysis continues

(see Diaconescu et al., 2015; Li and Heap, 2014; Kotlarski et al., 2014). By doing so, it is advisable to recognise the numerical solver of the models: in case of discrete differences, grid cell values are representing averages throughout the grid cell, because of the underlying Reynolds averaging. In such case, a conservative remapping guarantees comparability. b) An intrinsic incomparability with IWV data from GPS stations is introduced, because GPS IWV is based on profiles from 4 surrounding ERA-Interim grid cells that are bi-linearly interpolated to the location (latitude and longitude) of the GPS station (Parracho et al., 2018). Hence, the effective resolution of the GPS IWV data is much lower than all models in the manuscript – it is even lower than the IWV from ERA-Interim.

(2) Internal variability and its influence on evaluation results The solution of a local area model is partly predominated by its lateral boundary conditions (LBCs). The larger the model domain or the smaller the grid spacing becomes, the weaker becomes the coupling to its LBCs and the larger become large-scale deviations from its driving data in the interior of the model. Kida et al. (1991) and Paegle et al. (1996) are often cited in this context. More recently, Becker et al. (2015) demonstrated that a local area model creates artificial flows to compensate those deviations in order to achieve physical consistency with the LBCs along the lateral boundaries and that an increase of the model domain does not change this – the artificial flows simply become more complex. As a consequence of this decoupling the model's variability is increased compared to its driving data. This may lead to added value if the LBCs are derived from a global climate model. However, if the LBCs are taken from ERA-Interim (or some other reanalysis product), the decoupling introduces deviations from observational data. Such deviations are not "wrong", they just limit the applicability of traditional error statistics. For instance, if there is a thunderstorm at a certain point in time at a certain location in the observations, one cannot expect to find the same thunderstorm at the same location in the model. This has a severe impact on biases that are calculated grid cell by grid cell, but it does not mean, that the model is "wrong" – in a climatological context. This decoupling effect can be seen for instance in Table 3: SD from daily differences

are systematically smaller than SD from 6 hourly data and correlation coefficients on monthly basis are very – although these numbers are affected by issue a). All biases from IPSL20 are systematically larger than those from IPSL50, although both simulations are nudged to large-scale dynamics.

Both methodological issues (comparability and internal variability) have not received any attention yet. However, these issues may severely contribute to the detected biases and their interpretations (which are numerous throughout the manuscript) and hence, they could have significant impact on the conclusions. The authors are kindly asked to revise their analyses, interpretations, and conclusions according to the suggestions below. In order to achieve comparability, all model data should be remapped onto a common grid first and continue with the analysis afterwards. This common grid may depend on the variable, the models, and the reference data. For instance, if IWV from models need to be compared with IWV from GPS, then all models need to be remapped onto the ERA-Interim grid; the final IWV is then derived by a bi-linear interpolation from the 4 surrounding grid cells. In order to avoid misinterpretations of biases stemming from internal variability, one can increase the period and/or the area of averaging. At least, the interpretation of 6 hourly biases should be avoided.

Becker, N., Ulbrich, U. and Klein, R.: Systematic large-scale secondary circulations in a regional climate model, Geophys. Res. Lett., 42(10), 4142–4149, doi:10.1002/2015GL063955, 2015. Diaconescu, E. P., Gachon, P. and Laprise, R.: On the Remapping Procedure of Daily Precipitation Statistics and Indices Used in Regional Climate Model Evaluation, J. Hydrometeorol., 16(6), 2301–2310, doi:10.1175/JHM-D-15-0025.1, 2015. Kida, H., Koide, T., Sasaki, H. and Chiba, M.: A New Approach for Coupling a Limited Area Model to a Gcm for Regional Climate Simulations, J. Meteorol. Soc. Jpn., 69(6), 723–728, 1991. Kotlarski, S., Keuler, K., Christensen, O. B., Colette, A., Déqué, M., Gobiet, A., Goergen, K., Jacob, D., Lüthi, D., van Meijgaard, E., Nikulin, G., Schär, C., Teichmann, C., Vautard, R., Warrach-Sagi, K. and Wulfmeyer, V.: Regional climate modeling on European scales: a joint standard evaluation of the EURO-CORDEX RCM ensemble, Geosci. Model Dev., 7(4), 1297–1333, doi:10.5194/gmd-7-1297-2014, 2014. Li, J. and Heap, A. D.: Spatial interpolation methods applied in the environmental sciences: A review, Environ. Model. Softw., 53, 173–189, doi:10.1016/j.envsoft.2013.12.008, 2014. Paegle, J., Mo, K. and NoguesPaegle, J.: Dependence of simulated precipitation on surface evaporation during the 1993 United States summer floods, Mon. Weather Rev., 124(3), 345–361, doi:10.1175/1520-0493(1996)124<0345:DOSPOS>2.0.CO;2, 1996. Parracho, A. C., Bock, O. and Bastin, S.: Global IWV trends and variability in atmospheric reanalyses and GPS observations, Atmospheric Chem. Phys. Discuss., 1–43, doi:10.5194/acp-2018-137, 2018.

Specific Comments

Page 3, line 38: Parracho et al. (2018) only speaks of 104 GPS stations world wide. How can the authors make use of a hundred of European sites? Page 3, line 39: How accurate are such IWV measurements in the end? Page 5, line 18: ERA-Interim should be referred by Dee et al. (2011). Dee, D. P., Uppala, S. M., Simmons, A. J., Berrisford, P., Poli, P., Kobayashi, S., Andrae, U., Balmaseda, M. A., Balsamo, G., Bauer, P., Bechtold, P., Beljaars, A. C. M., van de Berg, L., Bidlot, J., Bormann, N., Delsol, C., Dragani, R., Fuentes, M., Geer, A. J., Haimberger, L., Healy, S. B., Hersbach, H., Holm, E. V., Isaksen, L., Kallberg, P., Koehler, M., Matricardi, M., McNally, A. P., Monge-Sanz, B. M., Morcrette, J.-J., Park, B.-K., Peubey, C., de Rosnay, P., Tavolato, C., Thepaut, J.-N. and Vitart, F.: The ERA-Interim reanalysis: configuration and performance of the data assimilation system, Q. J. R. Meteorol. Soc., 137(656), 553–597, doi:10.1002/qj.828, 2011. Page 6ff: When models are compared with observational data, the authors simply speak of "differences". However, it is not always clear how the difference is defined: "model minus observation" (as I suppose) or "observation minus model"? Please, define "difference" somewhere in the methods section and stay with it throughout the manuscript. Page 7, line 39: The explanation of the temperature binning should be explained here. Why is it important that all bins have a similar amount of

data elements? Temperatures do not occur with the same frequency – in fact, it would be easier to follow the argumentation, if the binning would be the same for all models and observations. Page 14, line 29: "The humidity bias thus strongly affects the low precipitation rates, more than the threshold of precipitation triggering." The latter part of this sentence is inconclusive: 1) a ranking of possible reasons has not been done – however, it would be nice to have. Maybe the authors could explicitly work out this point. 2) Which threshold is meant in this context?

Technical Corrections

The authors are introducing a space character (" ") prior to a double point (":"). I find this quite disturbing. It would help, if these unusual space characters could be avoided. Sometimes the LMDZ model is labelled with "LMD", sometimes it is labelled with "LMD50". Just for the sake of consistency and also to provide some information about the grid spacing in the acronym, I suggest to use "LMD50" throughout the manuscript. Page 1: line 29: typo: "baises" Page 6, line 23: typo: "Various evaluation metrics . . . has . . ." Page 7, line 39: referring to Figure 7 at this stage is way too early. The numeration of figures and tables should follow the sequence of their first appearance. Page 8, line 10: "This one identifies the minimum value . . ." – shouldn't it be the maximum? Page 9, line 23: The sentence "This good agreement . . ." is speculative and not relevant for the presented work. Page 9, line 34: typo: ". . . is a very godd approximation. . ." Page 10, line 18: "Table 5" – sequence of numeration Page 10, line 18: typo: ". . . averaged valuers . . ." Page 10, line 29: " . . . addects . . ." – not clear, what is meant by that; maybe "dominates"? Page 12, line 13: typo " . . . depiste . . ." Page 12, line 16: typo " . . . aslo . . ." Page 12, line 33: " . . . explain important SD . . ." – not clear, what is meant by "important"; maybe "large parts"? Page 13, line 5: ". . . precipitation picks up . . ." – this phrase is often used in the manuscript. It sounds a bit clumsy. Precipitation more likely "starts to increase". Page 13, line 11: typo " . . . the same than . . ." Page 13, line 29: typo " . . . tendancy . . ." Page 23, Table 3: Are the numbers differences in IWV (as I suppose)? Figure 1, 5c, and 6c: I suggest to

reverse the colours for IWV. When it is more moist (large values) it should be blue. Figure 2: The labels "ReOBS", "COM 1pc", and "COM maille" are not defined, here. In more general, it would increase the readability, if the legend and the figure caption would make use of the same acronyms. Figure 2d: Is there a reason for including non-precipitating days in q50? If there are more non-precipitating days than precipitating days q50 simply becomes 0. It would be more informative, if q50 would be based on precipitating days only. Figure 5d: It would be more informative, if Tb1 and Tb2 would be indicated. Figure 6a: What is that red dashed line?
* * *

---

## Author Comment (AC1) · 14 Dec 2018

A. Referee #1 comments and authors' response on "Impact of humidity biases on light precipitation occurrence: observations versus simulations" J.J. Gómez-Navarro (Referee) jjgomeznavarro@um.es

1 Abstract This study evaluates the performance of a number of regional climate models to reproduce humidity and precipitation. The emphasis of the article lies on the evaluation of Integrated Water Vapor (IWV), as well as its relation with temperature and finally with precipitation. It is found that models tend to overestimate the lower values of IWV, which is closely related to the "drizzling effect", i.e. too often too low

precipitation.

2 General comments I find this an interesting piece of work. The authors have made an effort to collect and put in comparable terms an heterogeneous set of data, from models to different observed variables at different locations and with different temporal availability. The use of an ensemble of simulations is a particularly good choice, as it provides robustness in the findings. I also like the fact that the authors do not just compare models with observations in plain terms, but they describe a simple two-layer atmosphere model that allows them to modelise the relationship between IWV and relative humidity with temperature, which allows them to gain insight on the sources of model biases. The text is well written and is easy to read, and the conclusions follow from the analysis carried out. Therefore I have only found very minor issues that the authors might want to consider.

»We thank the referee for the positive comments and we took in consideration all the minor issues listed in section 3.

3 Specific comments & Technical corrections 1. Pag. 6, line 23: has been → have been -> done 2. Pag. 7, line 1: consists in → consists of -> done 3. Pag. 7, line 34: Due to the existence of gaps in the observational dataset, which reduces... -> done 4. Pag. 10, line 18: valeurs → values? -> done 5. Pag. 11, line 20: I'd suggest that s, in Qs(T) to be called subscript, not exponent. An exponent is something else -> done 6. Pag 10, line 28: Why do we have that Qs(TFT)≈$\alpha$Qs(TBL)? It is not obvious to me.

»The August-Roche-Magnus formula (approximation of Clausius-Clapeyron law) allows to express qsat (specific humidity at saturation) in function of Ta (air temperature (in °C)) :

es = 6.1094*exp((17.625*Ta)./(243.04+Ta)); qsat = 0.622*(es./(Pa-es));

Fig.AC1toRC1 plots qsat = f(Ta) and it shows that for low T, qsat is less than 2 g/kg, and it increases exponentially at high T. But in our 2-layers model, we consider the averaged

temperature of the boundary layer, not the surface temperature, so it is not so high, and we consider that T is around 280-290K, so qsat is between 6 and 12 g/kg ; and for the free troposphere, the temperature is less than 260K and qsat is less than 2 g/kg. The ratio between Qs(TBL) and Qs(TFT) is then close to $1/\alpha$. In the text, we added some indications on these values and we mentionned the August-Magnus-Roche formula.

7. Pag. 12, line 8: What is Tb1 and Tb2? »Tb1 is the temperature after which RH starts to decrease, while Tb2 is the temperature after which IWV starts to decrease. We clarified that in the text in this way :

'RH starts to decrease significantly at T~13°C (hereafter called Tb1), while IWV curve deflects at T~16°C (hereafter called Tb2)'

We also added vertical lines on Fig. 5b to indicate Tb1 and Tb2.

8. Pag. 12, line 33: What is SD? »SD is for standard deviation. We replaced SD by the full name.

9. Figure 5: Should the right panel in the first row be labelled d? Further, that panel has particularly low resolution and generally lower presentation quality than, let's say, Figure 4

» the right panel of the first row is now labeled b, and the two plots under it are now c and d. References to these subplots have been modified in the text accordingly. We also improved the quality of this figure and added vertical lines for Tb1 and Tb2 on subplot b. The colors for IWV have also been inverted to have hot colors for dry areas and cold colors for humid ones, as suggested by reviewer 2.

10. Figure 6: The same applies here. The first panel has bad quality and different aesthetics. I'd advice to follow the style followed to produce Fig. 4

»We tried to improve the esthetics of the figure as much as we could.

11. Figure 7: the lines surrounding the panels are partly hidden by them and produce

an ugly effect that should be avoided in the final version of the manuscript

» Figure 7 now appears correctly in the PDF file.

12. Figure 9: The panels could be larger to take better advantage of the available space. Unlike in the other two, the third column has no right and top axis. The labels in the first row overlap with the axes. The low resolution issue applies here as well. There is no legend.

» We added a legend and right and top axis in the third column. Labels are also better located and resolution has been improved.

The marked-up manuscript including all the changes has been uploaded as *.pdf supplement

Please also note the supplement to this comment:
https://www.atmos-chem-phys-discuss.net/acp-2018-624/acp-2018-624-AC1-supplement.pdf

[Figure]

**Fig. 1.** Fig.AC1toRC1:Specific humidity at saturation in function of temperature following August-Magnus-Roche approximation of Clausius-Clapeyron law.

---

## Author Comment (AC2) · 14 Dec 2018

We are very grateful to referee 2 for its positive and relevant comments which allowed to address important issues in our revised manuscript. We took into account all the comments as follow :

The authors make use of integrated water vapour (IWV) and precipitation from a col-

lection of observations (GPS stations, radio sounding, and combined radar/rain gauge data) in order to evaluate regional climate models operated on various grids (the grid spacing ranges from 0.11◦ to 0.44◦) in a climatological manner (the periods of evaluation cover multiple years to decades). The climate models' simulations are driven by ERA-Interim and participated in the Med-CORDEX initiative. For this purpose, the authors develop a conceptual model that connects IWV, temperature, and precipitation (including Clausius-Clapeyron scaling and deviations from it) that helps to interpret the detected model biases and gives insights in the complexity of precipitation generating processes. From their analyses the authors conclude: 1) all models overestimate lower values of IWV with an increasing spread among the models during summertime 2) mean biases are mostly explained by model physics (land surface/atmosphere interactions) while dynamics affect the variability 3) the IWV/temperature relationship (that deviates from the Clausius-Clapeyron law) is generally well represented by the models 4) biases in the frequency of occurrence in precipitation can be explained by a higher probability of exceedance of a critical value for IWV (that in turn depends on temperature)

General Comments There is an endless number of evaluation papers for regional climate models (RCMs) that content themselves with showing biases, but there are only a few papers that deal with the sources of such biases and their underlying processes. The presented manuscript could be one of those rare papers. In addition, the manuscript elaborates on (even if just in a speculative manner) some aspects of the question, how climate change may affect the water cycle. The authors have provided a very interesting and innovative analysis that should be published as soon as possible. However, there are two methodological weaknesses that should to be clarified first, because they may affect the conclusions concerning the interpretation of biases and the precipitation/IWV function (Figure 7) drawn: (1) Comparability between reference (observational) and modelled data The authors make use of RCM data (including re-analysis data ERA-Interim) on various grids and compare it with data from stations (point data) by means of the nearest neighbouring method (cf. page 6, line 12). This

has two implications: a) In such a comparison a coarser resolved model is more penalised than a model with a higher resolution, because it has a smaller spatial variability per construction. The coarser resolved model "sees" processes that are resolved by the model with a higher resolution only as "sub-grid scale effects". As a consequence, a judgement of biases drawn from models on their original grid can be misleading. In order to achieve comparability throughout models of different resolution, modelled and observed data is usually remapped onto a common coarsely resolved grid before the analysis continues (see Diaconescu et al., 2015; Li and Heap, 2014; Kotlarski et al., 2014). By doing so, it is advisable to recognise the numerical solver of the models: in case of discrete differences, grid cell values are representing averages throughout the grid cell, because of the underlying Reynolds averaging. In such case, a conservative remapping guarantees comparability. b) An intrinsic incomparability with IWV data from GPS stations is introduced, because GPS IWV is based on profiles from 4 surrounding ERA-Interim grid cells that are bi-linearly interpolated to the location (latitude and longitude) of the GPS station (Parracho et al., 2018). Hence, the effective resolution of the GPS IWV data is much lower than all models in the manuscript – it is even lower than the IWV from ERA-Interim.

-> We draw the attention of the referee to the fact that the profile data from the 4 surrounding ERA-Interim grid cells used in the computation of the GPS IWV data are: 1) an estimate of the surface pressure Ps, and 2) an estimate of the weighted mean temperature Tm, but not of IWV. Hence we believe that the water vapour information contained in the GPS ZTD data remain representative of the atmospheric column above the GPS antenna. Note that in Parracho et al. 2018, a bi-linearly interpolated IWV estimate is also computed from the 4 surrounding ERA-Interim grid cells but this one is used as the ERA-Interim IWV estimate. In the present work, ERA-Interim IWV data was not computed in the same way, but simply extracted from the nearest grid cell. It is therefore not necessary to derive the model IWV by a bi-linearly. However, we agree that mapping all models onto the ERA-Interim grid is a proper approach to compare all the models with GPS and we did it to compute the bias and standard deviation (Table

3), and the percentage of values which overestimate GPS values (Figure 3), and the relationship between monthly IWV values and surface humidity values (Figure 4).

-> For precipitation, we partly addressed this point in our comparison since we used the COMEPHORE product at two different resolutions: at 1 km resolution to compare it with the local measurement at SIRTA, and at 50km resolution to compare it with the models using a similar size of grid cell (see section 3.2). However, we agree that it is not fair to compare models at 50 km resolution and models at higher resolution. So, to address this point in our review, i) we removed all the simulations which use higher resolution than 50 km from our analysis because it is not the important message of this paper. ii) to compare the occurrence of precipitation and precipitation fluxes (Figure 2), we regridded the precipitation model outputs to the LMD grid using conservative remapping. But for the main analysis of this paper which aims at estimating the critical value of IWV from which precipitation starts to increase, we preferred to keep the native grid of the models so that it really corresponds to an intrinsic property of the model not modified by numerical artefact due to remapping.

(2) Internal variability and its influence on evaluation results The solution of a local area model is partly predominated by its lateral boundary conditions (LBCs). The larger the model domain or the smaller the grid spacing becomes, the weaker becomes the coupling to its LBCs and the larger become large-scale deviations from its driving data in the interior of the model. Kida et al. (1991) and Paegle et al. (1996) are often cited in this context. More recently, Becker et al. (2015) demonstrated that a local area model creates artificial flows to compensate those deviations in order to achieve physical consistency with the LBCs along the lateral boundaries and that an increase of the model domain does not change this – the artificial flows simply become more complex. As a consequence of this decoupling the model's variability is increased compared to its driving data. This may lead to added value if the LBCs are derived from a global climate model. However, if the LBCs are taken from ERA-Interim (or some other re-analysis product), the decoupling introduces deviations from observational data. Such

deviations are not "wrong", they just limit the applicability of traditional error statistics. For instance, if there is a thunderstorm at a certain point in time at a certain location in the observations, one cannot expect to find the same thunderstorm at the same location in the model. This has a severe impact on biases that are calculated grid cell by grid cell, but it does not mean, that the model is "wrong" – in a climatological context. This decoupling effect can be seen for instance in Table 3: SD from daily differences are systematically smaller than SD from 6 hourly data and correlation coefficients on monthly basis are very – although these numbers are affected by issue a). All biases from IPSL20 are systematically larger than those from IPSL50, although both simulations are nudged to large-scale dynamics.

-> The reviewer is right, the internal variability affects the results and we should not present results at 6-hourly time scales. So we removed them from our analysis. However, we think that we've partly addressed the internal variability aspect in the submitted version, by comparing the statistics obtained on precipitation occurrence for different periods (see Table 4), and by estimating the critical value of IWV using also different periods, and in particular the longer common period we have for the models that is 20-year long (1989-2008). These results were discussed in section 5.1, and presented in Fig. S1 of the submitted paper. In the revised version, we removed the comparison at 6-hourly and we discussed a bit more the issue of internal variability in section 4.1 when we compare IWV from GPS, ERAi and RCMs, using references suggested by the reviewer and others. We kept the discussion in section 5.1 which adresses the impact of the different periods.

Both methodological issues (comparability and internal variability) have not received any attention yet. However, these issues may severely contribute to the detected biases and their interpretations (which are numerous throughout the manuscript) and hence, they could have significant impact on the conclusions. The authors are kindly asked to revise their analyses, interpretations, and conclusions according to the suggestions below. In order to achieve comparability, all model data should be remapped onto a

common grid first and continue with the analysis afterwards. This common grid may depend on the variable, the models, and the reference data. For instance, if IWV from models need to be compared with IWV from GPS, then all models need to be remapped onto the ERA-Interim grid; the final IWV is then derived by a bi-linear interpolation from the 4 surrounding grid cells. In order to avoid misinterpretations of biases stemming from internal variability, one can increase the period and/or the area of averaging. At least, the interpretation of 6 hourly biases should be avoided.

-> See previous comments. We remapped the outputs to a common grid (ERAI grid when comparing IWV datasets, and LMD for precipitation occurrence) and we removed higher resolution simulations and 6-hourly datasets from our analysis.

Becker, N., Ulbrich, U. and Klein, R.: Systematic large-scale secondary circulations in a regional climate model, Geophys.Res.Lett., 42(10), 4142–4149, doi:10.1002/2015GL063955, 2015. Diaconescu, E. P., Gachon, P. and Laprise, R.: On the Remapping Procedure of Daily Precipitation Statistics and Indices Used in Regional Climate Model Evaluation, J. Hydrometeorol., 16(6), 2301–2310, doi:10.1175/JHM-D-15-0025.1, 2015. Kida, H., Koide, T., Sasaki, H. and Chiba, M.: A New Approach for Coupling a Limited Area Model to a Gcm for Regional Climate Simulations, J. Meteorol. Soc. Jpn., 69(6), 723–728, 1991. Kotlarski, S., Keuler, K., Christensen, O. B., Colette, A., Déqué, M., Gobiet, A., Goergen, K., Jacob, D., Lüthi, D., van Meijgaard, E., Nikulin, G., Schär, C., Teichmann, C., Vautard, R., Warrach-Sagi, K. and Wulfmeyer, V.: Regional climate modeling on European scales: a joint standard evaluation of the EURO-CORDEX RCM ensemble, Geosci. Model Dev., 7(4), 1297–1333, doi:10.5194/gmd-7-1297-2014, 2014. Li, J. and Heap, A. D.: Spatial interpolation methods applied in the environmental sciences: A review, Environ. Model. Softw., 53, 173–189, doi:10.1016/j.envsoft.2013.12.008, 2014. Paegle, J., Mo, K. and NoguesPaegle, J.: Dependence of simulated precipitation on surface evaporation during the 1993 United States summer floods, Mon. Weather Rev., 124(3), 345–361, doi:10.1175/1520-0493(1996)124<0345:DOSPOS>2.0.CO;2, 1996. Parracho, A. C.,

Bock, O. and Bastin, S.: Global IWV trends and variability in atmospheric reanalyses and GPS observations, Atmospheric Chem. Phys. Discuss., 1–43, doi:10.5194/acp-2018-137, 2018.

Specific Comments Page 3, line 38: Parracho et al. (2018) only speaks of 104 GPS stations world wide. How can the authors make use of a hundred of European sites?

-> In Parracho et al. (2018), as explained in their section "2.2 GPS data", among 456 GPS stations, stations that have time series with only small gaps over the 15-year periods 1995-2010 have been selected so that they could estimate trends. In our study, we are not as restrictive as they are because we do not estimate trends. We selected stations with at least 5 years of available data. We modified this sentence to make clearer the fact that the processing is the same than those used by Parracho et al., but not the selection of stations. We didn't give too much details in the introduction but we gave more details in the "Material" section (section 2.1).

Page 3, line 39: How accurate are such IWV measurements in the end?

-> The accuracy of GPS IWV has been estimated about 1-2 kg/m2 (Bock et al., 2005; 2013; Ning et al., 2016).

Bock O, Keil C, Richard E, Flamant C, Bouin MN. Validation of precipitable water from ECMWF model analyses with GPS and radiosonde data during the MAP SOP. Q. J. R. Meteorol. Soc. 131:3013–3036, 2005 Bock, O., Bosser, P., Bourcy, T., David, L., Goutail, F., Hoareau, C., Keckhut, P., Legain, D., Pazmino, A., Pelon, J., Pipis, K., Poujol, G., Sarkissian, A., Thom, C., Tournois, G., and Tzanos, D.: Accuracy assessment of water vapour measurements from in situ and remote sensing techniques during the DEMEVAP 2011 campaign at OHP, Atmos. Meas. Tech., 6, 2777-2802, https://doi.org/10.5194/amt-6-2777-2013, 2013. Ning, T., Wang, J., Elgered, G., Dick, G., Wickert, J., Bradke, M., Sommer, M., Querel, R., and Smale, D.: The uncertainty of the atmospheric integrated water vapour estimated from GNSS observations, Atmos. Meas. Tech., 9, 79-92, https://doi.org/10.5194/amt-9-79-2016, 2016.

Page 5, line 18: ERA-Interim should be referred by Dee et al. (2011). Dee, D. P., Uppala, S. M., Simmons, A. J., Berrisford, P., Poli, P., Kobayashi, S., Andrae, U., Balmaseda, M. A., Balsamo, G., Bauer, P., Bechtold, P., Beljaars, A. C. M., van de Berg, L., Bidlot, J., Bormann, N., Delsol, C., Dragani, R., Fuentes, M., Geer, A. J., Haimberger, L., Healy, S. B., Hersbach, H., Holm, E. V., Isaksen, L., Kallberg, P., Koehler, M., Matricardi, M., McNally, A. P., Monge-Sanz, B. M., Morcrette, J.-J., Park, B.-K., Peubey, C., de Rosnay, P., Tavolato, C., Thepaut, J.-N. and Vitart, F.: The ERA Interim reanalysis: configuration and performance of the data assimilation system, Q. J. R. Meteorol. Soc., 137(656), 553–597,doi:10.1002/qj.828, 2011.

-> done: we replaced reference to Berrisford et al., 2011 by this reference.

Page 6ff: When models are compared with observational data, the authors simply speak of "differences". However, it is not always clear how the difference is defined: "model minus observation" (as I suppose) or "observation minus model"? Please, define "difference" somewhere in the methods section and stay with it throughout the manuscript.

->We added this sentence at the beginning of the 'Methods' section and we checked that it is like this in our study. "To compare models and observations, we consider differences as the 'model minus observation' results throughout the manuscript."

Page 7, line 39: The explanation of the temperature binning should be explained here. Why is it important that all bins have a similar amount of data elements? Temperatures do not occur with the same frequency – in fact, it would be easier to follow the argumentation, if the binning would be the same for all models and observations.

-> Our methodology ensures a reasonable number of events in all bins. However, you're right that since we do not consider extremes, and we do not focus on lower or upper tail of the distributions, we can use the same bins for temperature in models and observations. In the revised version, we changed the text and the results according to that.

Page 14, line 29: "The humidity bias thus strongly affects the low precipitation rates, more than the threshold of precipitation triggering." The latter part of this sentence is inconclusive: 1) a ranking of possible reasons has not been done – however, it would be nice to have. Maybe the authors could explicitly work out this point. 2) Which threshold is meant in this context?

-> We removed this sentence and added this point as a perspective in the conclusion.

Technical Corrections The authors are introducing a space character (" ") prior to a double point (":"). I find this quite disturbing. It would help, if these unusual space characters could be avoided.

-> we removed the space character prior to double point in all the documents. It seems that this space is created automatically by Word with the police "Times New Roman".

Sometimes the LMDZ model is labelled with "LMD", sometimes it is labelled with "LMD50". Just for the sake of consistency and also to provide some information about the grid spacing in the acronym, I suggest to use "LMD50" throughout the manuscript.

-> Reviewer is right. We used LMD50 throughout the manuscript.

Page 1: line 29: typo: "baises"

-> done

Page 6, line 23: typo: "Various evaluation metrics ...has..."

-> done

Page 7, line 39: referring to Figure 7 at this stage is way too early. The numeration of figures and tables should follow the sequence of their first appearance.

-> we removed the reference to this figure at this stage.

Page 8, line 10: "This one identifies the minimum value..." – shouldn't it be the maximum?

-> Actually, we modified this sentence to better take into account the uncertainty in the estimate of this value. It is now this one: "This one identifies the value(s) of IWV over which precipitation rates are greater than 0.1 mm/day. In some cases, two values are obtained, that are represented by an errorbar to indicate the uncertainty of the estimate of this critical value. Âż These errorbars appear on Fig.8

Page 9, line 23: The sentence "This good agreement..." is speculative and not relevant for the presented work.

-> we removed this sentence

Page 9, line 34: typo: "...is a very godd approximation..."

-> done

Page 10, line 18: "Table 5" – sequence of numeration

-> we checked that the sequence of numeration is ok

Page 10, line 18: typo: "...averaged valuers..."

-> done

Page 10, line 29: "...addects..." – not clear, what is meant by that; maybe "dominates"?

-> it's a mistake. We replaced 'addects' by 'affects'

Page 12, line 13: typo "...depiste..."

-> done

Page 12, line 16: typo "...aslo..."

-> done

Page 12, line 33: "...explain important SD..."– not clear, what is meant by "important"; maybe "large parts"?

[Figure]

-> we used reviewer's suggestion to replace 'important'

Page 13, line 5: "...precipitation picks up..." – this phrase is often used in the manuscript. It sounds a bit clumsy. Precipitation more likely "starts to increase".

-> done. We replaced 'pick up' by 'start to increase'.

Page 13, line 11: typo "...the same than..."

-> done

Page 13, line 29: typo "...tendancy..."

-> done

Page 23, Table 3: Are the numbers differences in IWV (as I suppose)?

-> we added IWV in the caption.

Figure 1, 5c, and 6c: I suggest to reverse the colours for IWV. When it is more moist (large values) it should be blue.

-> done

Figure 2: The labels "ReOBS", "COM 1pc", and "COM maille" are not defined, here. In more general, it would increase the readability, if the legend and the figure caption would make use of the same acronyms.

-> we added the labels in the caption.

Figure 2d: Is there a reason for including non-precipitating days in q50? If there are more non-precipitating days than precipitating days q50 simply becomes 0. It would be more informative, if q50 would be based on precipitating days only.

-> reviewer is right, it is redondant with Fig.2a. In the revised version, we based q50 on precipitating days only.

Figure 5d: It would be more informative, if Tb1 and Tb2 would be indicated.

-> done

Figure 6a: What is that red dashed line?

-> it shows the Clausius-Clapeyron scaling. It is now indicated in the caption.

The marked-up manuscript including all our changes in manuscript has been uploaded as a *.pdf supplement.

Please also note the supplement to this comment:
https://www.atmos-chem-phys-discuss.net/acp-2018-624/acp-2018-624-AC2-supplement.pdf
* * *
[Figure]

**Supplement:**

B. Marked-up manuscript with authors' changes

[revised manuscript text omitted]